# Recursive Sampling for the Nyström Method

**Cameron Musco**
MIT EECS
cnmusco@mit.edu

**Christopher Musco**
MIT EECS
cpmusco@mit.edu

## Abstract

We give the first algorithm for kernel Nyström approximation that runs in *linear time in the number of training points* and is provably accurate for all kernel matrices, without dependence on regularity or incoherence conditions. The algorithm projects the kernel onto a set of $s$ landmark points sampled by their *ridge leverage scores*, requiring just $O(ns)$ kernel evaluations and $O(ns^2)$ additional runtime. While leverage score sampling has long been known to give strong theoretical guarantees for Nyström approximation, by employing a fast recursive sampling scheme, our algorithm is the first to make the approach scalable. Empirically we show that it finds more accurate kernel approximations in less time than popular techniques such as classic Nyström approximation and the random Fourier features method.

## 1 Introduction

The kernel method is a powerful for applying linear learning algorithms (SVMs, linear regression, etc.) to nonlinear problems. The key idea is to map data to a higher dimensional *kernel feature space*, where linear relationships correspond to nonlinear relationships in the original data.

Typically this mapping is implicit. A *kernel function* is used to compute inner products in the high-dimensional kernel space, without ever actually mapping original data points to the space. Given $n$ data points $\mathbf{x}_1, \ldots, \mathbf{x}_n$, the $n \times n$ kernel matrix $\mathbf{K}$ is formed where $\mathbf{K}_{i,j}$ contains the high-dimensional inner product between $\mathbf{x}_i$ and $\mathbf{x}_j$, as computed by the kernel function. All computations required by a linear learning method are performed using the inner product information in $\mathbf{K}$.

Unfortunately, the transition from linear to nonlinear comes at a high cost. Just generating the entries of $\mathbf{K}$ requires $\Theta(n^2)$ time, which is prohibitive for large datasets.

### 1.1 Kernel approximation

A large body of work seeks to accelerate kernel methods by finding a compressed, often low-rank, approximation $\tilde{\mathbf{K}}$ to the true kernel matrix $\mathbf{K}$. Techniques include random sampling and embedding [AMS01, BBV06, ANW14], *random Fourier feature* methods for shift invariant kernels [RR07, RR09, LSS13], and incomplete Cholesky factorization [FS02, BJ02].

One of the most popular techniques is the *Nyström method*, which constructs $\tilde{\mathbf{K}}$ using a subset of "landmark" data points [WS01]. Once $s$ data points are selected, $\tilde{\mathbf{K}}$ (in factored form) takes just $O(ns)$ kernel evaluations and $O(s^3)$ additional time to compute, requires $O(ns)$ space to store, and can be manipulated quickly in downstream applications. E.g., inverting $\tilde{\mathbf{K}}$ takes $O(ns^2)$ time.

The Nyström method performs well in practice [YLM+12, GM13, TRVR16], is widely implemented [HFH+09, PVG+11, IBM14], and is used in a number of applications under different names such as "landmark isomap" [DST03] and "landmark MDS" [Pla05]. In the classic variant, landmark points are selected uniformly at random. However, significant research seeks to improve performance via data-

dependent sampling that selects landmarks which more closely approximate the full kernel matrix than uniformly sampled landmarks [SS00, DM05, ZTK08, BW09, KMT12, WZ13, GM13, LJS16].

Theoretical work has converged on *leverage score* based approaches, as they give the strongest provable guarantees for both kernel approximation [DMM08, GM13] and statistical performance in downstream applications [AM15, RCR15, Wan16]. Leverage scores capture how important an individual data point is in composing the span of the kernel matrix.

Unfortunately, these scores are prohibitively expensive to compute. All known approximation schemes require $\Omega(n^2)$ time or only run quickly under strong conditions on $\mathbf{K}$ – e.g. good conditioning or data "incoherence" [DMIMW12, GM13, AM15, CLV16]. Hence, leverage score-based approaches remain largely in the domain of theory, with limited practical impact [KMT12, LBKL15, YPW15].

## 1.2 Our contributions

In this work, we close the gap between strong approximation bounds and efficiency: we present a new Nyström algorithm based on *recursive leverage score sampling* which achieves the "best of both worlds": it produces kernel approximations provably matching the high accuracy of leverage score methods while only requiring $O(ns)$ kernel evaluations and $O(ns^2)$ runtime for $s$ landmark points.

Theoretically, this runtime is surprising. In the typical case when $s \ll n$, the algorithm evaluates just a small subset of $\mathbf{K}$, ignoring most of the kernel space inner products. Yet its performance guarantees hold for general kernels, requiring *no assumptions on coherence or regularity*.

Empirically, the runtime's linear dependence on $n$ means that our method is the first leverage score algorithm that can compete with the most commonly implemented techniques, including the classic uniform sampling Nyström method and random Fourier features sampling [RR07]. Since our algorithm obtains higher quality samples, we show experimentally that it outperforms these methods on benchmark datasets – it can obtain as accurate a kernel approximation in significantly less time. Our approximations also have lower rank, so they can be stored in less space and processed more quickly in downstream learning tasks.

## 1.3 Paper outline

Our recursive sampling algorithm is built on top of a Nyström scheme of Alaoui and Mahoney that samples landmark points based on their *ridge leverage scores* [AM15]. After reviewing preliminaries in Section 2, in Section 3 we analyze this scheme, which we refer to as *RLS-Nyström*. To simplify prior work, which studies the statistical performance of RLS-Nyström for specific kernel learning tasks [AM15, RCR15, Wan16], we prove a strong, application independent approximation guarantee: for any $\lambda$, if $\tilde{\mathbf{K}}$ is constructed with $s = \Theta(d_{\text{eff}}^{\lambda} \log d_{\text{eff}}^{\lambda})$ samples[1], where $d_{\text{eff}}^{\lambda} = \text{tr}(\mathbf{K}(\mathbf{K} + \lambda\mathbf{I})^{-1})$ is the so-called "$\lambda$-effective dimensionality" of $\mathbf{K}$, then with high probability, $\|\mathbf{K} - \tilde{\mathbf{K}}\|_2 \leq \lambda$.

In Appendix E, we show that this guarantee implies bounds on the statistical performance of RLS-Nyström for kernel ridge regression and canonical correlation analysis. We also use it to prove new results on the performance of RLS-Nyström for kernel rank-$k$ PCA and $k$-means clustering – in both cases just $O(k \log k)$ samples are required to obtain a solution with good accuracy.

After affirming the favorable theoretical properties of RLS-Nyström, in Section 4 we show that its runtime can be significantly improved using a recursive sampling approach. Intuitively our algorithm is simple. We show how to approximate the kernel ridge leverage scores using a *uniform* sample of $\frac{1}{2}$ of our input points. While the subsampled kernel matrix still has a prohibitive $n^2/4$ entries, we can *recursively approximate* it, using our same sampling algorithm. If our final Nyström approximation will use $s$ landmarks, the recursive approximation only needs rank $O(s)$, which lets us estimate the ridge leverage scores of the original kernel matrix in just $O(ns^2)$ time. Since $n$ is cut in half at each level of recursion, our total runtime is $O\left(ns^2 + \frac{ns^2}{2} + \frac{ns^2}{4} + ...\right) = O(ns^2)$, significantly improving upon the method of [AM15], which takes $\Theta(n^3)$ time in the worst case.

Our approach builds on recent work on iterative sampling methods for approximate linear algebra [CLM$^+$15, CMM17]. While the analysis in the kernel setting is technical, our final algorithm is

simple and easy to implement. We present and test a parameter-free variation of Recursive RLS-Nyström in Section 5, confirming superior performance compared to existing methods.

## 2 Preliminaries

Consider an input space $\mathcal{X}$ and a positive semidefinite kernel function $K : \mathcal{X} \times \mathcal{X} \to \mathbb{R}$. Let $\mathcal{F}$ be an associated reproducing kernel Hilbert space and $\phi : \mathcal{X} \to \mathcal{F}$ be a (typically nonlinear) feature map such that for any $\mathbf{x}, \mathbf{y} \in \mathcal{X}$, $K(\mathbf{x}, \mathbf{y}) = \langle \phi(\mathbf{x}), \phi(\mathbf{y}) \rangle_{\mathcal{F}}$. Given a set of $n$ input points $\mathbf{x}_1, \ldots, \mathbf{x}_n \in \mathcal{X}$, define the kernel matrix $\mathbf{K} \in \mathbb{R}^{n \times n}$ by $\mathbf{K}_{i,j} = K(\mathbf{x}_i, \mathbf{x}_j)$.

It is often natural to consider the kernelized data matrix that generates $\mathbf{K}$. Informally, let $\mathbf{\Phi} \in \mathbb{R}^{n \times d'}$ be the matrix containing $\phi(\mathbf{x}_1), ..., \phi(\mathbf{x}_n)$ as its rows (note that $d'$ may be infinite). $\mathbf{K} = \mathbf{\Phi}\mathbf{\Phi}^T$. While we use $\mathbf{\Phi}$ for intuition, in our formal proofs we replace it with any matrix $\mathbf{B} \in \mathbb{R}^{n \times n}$ satisfying $\mathbf{B}\mathbf{B}^T = \mathbf{K}$ (e.g. a Cholesky factor). Such a $\mathbf{B}$ is guaranteed to exist since $\mathbf{K}$ is positive semidefinite.

We repeatedly use the singular value decomposition, which allows us to write any rank $r$ matrix $\mathbf{M} \in \mathbb{R}^{n \times d}$ as $\mathbf{M} = \mathbf{U}\mathbf{\Sigma}\mathbf{V}^{\mathbf{T}}$, where $\mathbf{U} \in \mathbb{R}^{n \times r}$ and $\mathbf{V} \in \mathbb{R}^{d \times r}$ have orthogonal columns (the left and right singular vectors of $\mathbf{M}$), and $\mathbf{\Sigma} \in \mathbb{R}^{r \times r}$ is a positive diagonal matrix containing the singular values: $\sigma_1(\mathbf{M}) \geq \sigma_2(\mathbf{M}) \geq \ldots \geq \sigma_r(\mathbf{M}) > 0$. $\mathbf{M}$'s pseudoinverse is given by $\mathbf{M}^+ = \mathbf{V}\mathbf{\Sigma}^{-1}\mathbf{U}^T$.

### 2.1 Nyström approximation

The Nyström method selects a subset of "landmark" points and uses them to construct a low-rank approximation to $\mathbf{K}$. Given a matrix $\mathbf{S} \in \mathbb{R}^{n \times s}$ that has a single entry in each column equal to $1$ so that $\mathbf{KS}$ is a subset of $s$ columns from $\mathbf{K}$, the associated Nyström approximation is:

$$\tilde{\mathbf{K}} = \mathbf{KS}(\mathbf{S}^T\mathbf{KS})^+\mathbf{S}^T\mathbf{K}. \tag{1}$$

$\tilde{\mathbf{K}}$ can be stored in $O(ns)$ space by separately storing $\mathbf{KS} \in \mathbb{R}^{n \times s}$ and $(\mathbf{S}^T\mathbf{KS})^+ \in \mathbb{R}^{s \times s}$. Furthermore, the factors can be computed using just $O(ns)$ evaluations of the kernel inner product to form $\mathbf{KS}$ and $O(s^3)$ time to compute $(\mathbf{S}^T\mathbf{KS})^+$. Typically $s \ll n$ so these costs are significantly lower than the cost to form and store the full kernel matrix $\mathbf{K}$.

We view Nyström approximation as a low-rank approximation to the dataset in feature space. Recalling that $\mathbf{K} = \mathbf{\Phi}\mathbf{\Phi}^T$, $\mathbf{S}$ selects $s$ kernelized data points $\mathbf{S}^T\mathbf{\Phi}$ and we approximate $\mathbf{\Phi}$ using its projection onto these points. Informally, let $\mathbf{P}_{\mathbf{S}} \in \mathbb{R}^{d' \times d'}$ be the orthogonal projection onto the row span of $\mathbf{S}^T\mathbf{\Phi}$. We approximate $\mathbf{\Phi}$ by $\tilde{\mathbf{\Phi}} \stackrel{\text{def}}{=} \mathbf{\Phi}\mathbf{P}_{\mathbf{S}}$. We can write $\mathbf{P}_{\mathbf{S}} = \mathbf{\Phi}^T\mathbf{S}(\mathbf{S}^T\mathbf{\Phi}\mathbf{\Phi}^T\mathbf{S})^+\mathbf{S}^T\mathbf{\Phi}$. Since it is an orthogonal projection, $\mathbf{P}_{\mathbf{S}}\mathbf{P}_{\mathbf{S}}^T = \mathbf{P}_{\mathbf{S}}^2 = \mathbf{P}_{\mathbf{S}}$, and so we can write:

$$\tilde{\mathbf{K}} = \tilde{\mathbf{\Phi}}\tilde{\mathbf{\Phi}}^T = \mathbf{\Phi}\mathbf{P}_{\mathbf{S}}^2\mathbf{\Phi}^T = \mathbf{\Phi}\left(\mathbf{\Phi}^T\mathbf{S}(\mathbf{S}^T\mathbf{\Phi}\mathbf{\Phi}^T\mathbf{S})^+\mathbf{S}^T\mathbf{\Phi}\right)\mathbf{\Phi}^T = \mathbf{KS}(\mathbf{S}^T\mathbf{KS})^+\mathbf{S}^T\mathbf{K}.$$

This recovers the standard Nyström approximation (1).

## 3 The RLS-Nyström method

We now introduce the RLS-Nyström method, which uses ridge leverage score sampling to select landmark data points, and discuss its strong approximation guarantees for any kernel matrix $\mathbf{K}$.

### 3.1 Ridge leverage scores

In classical Nyström approximation (1), $\mathbf{S}$ is formed by sampling data points uniformly at random. Uniform sampling can work in practice, but it only gives theoretical guarantees under strong regularity or incoherence assumptions on $\mathbf{K}$ [Git11]. It will fail for many natural kernel matrices where the relative "importance" of points is not uniform across the dataset

For example, imagine a dataset where points fall into several clusters, but one of the clusters is much larger than the rest. Uniform sampling will tend to oversample landmarks from the large cluster while undersampling or possibly missing smaller but still important clusters. Approximation of $\mathbf{K}$ and learning performance (e.g. classification accuracy) will decline as a result.

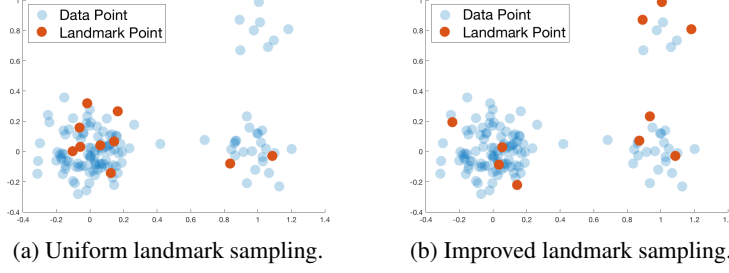

(a) Uniform landmark sampling.  (b) Improved landmark sampling.

Figure 1: Uniform sampling for Nyström approximation can oversample from denser parts of the dataset. A better Nyström scheme will select points that more equally cover the relevant data.

To combat this issue, alternative methods compute a measure of point importance that is used to select landmarks. For example, one heuristic applies $k$-means clustering to the input and takes the cluster centers as landmarks [ZTK08]. A large body of theoretical work measures importance using variations on the *statistical leverage scores*. One natural variation is the *ridge leverage score*:

**Definition 1** (Ridge leverage scores [AM15]). *For any $\lambda > 0$, the $\lambda$-ridge leverage score of data point $\mathbf{x}_i$ with respect to the kernel matrix $\mathbf{K}$ is defined as*

$$l_i^\lambda(\mathbf{K}) \stackrel{\text{def}}{=} \left(\mathbf{K}(\mathbf{K} + \lambda \mathbf{I})^{-1}\right)_{i,i}, \tag{2}$$

*where $\mathbf{I}$ is the $n \times n$ identity matrix. For any $\mathbf{B} \in \mathbb{R}^{n \times n}$ satisfying $\mathbf{B}\mathbf{B}^T = \mathbf{K}$, we can also write:*

$$l_i^\lambda(\mathbf{K}) = \mathbf{b}_i^T(\mathbf{B}^T\mathbf{B} + \lambda \mathbf{I})^{-1}\mathbf{b}_i, \tag{3}$$

*where $\mathbf{b}_i^T \in \mathbb{R}^{1 \times n}$ is the $i^{th}$ row of $\mathbf{B}$.*

For conciseness we typically write $l_i^\lambda(\mathbf{K})$ as $l_i^\lambda$. To check that (2) and (3) are equivalent note that $\mathbf{b}_i^T(\mathbf{B}^T\mathbf{B}+\lambda\mathbf{I})^{-1}\mathbf{b}_i = \left(\mathbf{B}(\mathbf{B}^T\mathbf{B} + \lambda\mathbf{I})^{-1}\mathbf{B}^T\right)_{i,i}$. Using the SVD to write $\mathbf{B} = \mathbf{U}\boldsymbol{\Sigma}\mathbf{V}^T$ and accordingly $\mathbf{K} = \mathbf{U}\boldsymbol{\Sigma}^2\mathbf{U}^T$ confirms that $\mathbf{K}(\mathbf{K}+\lambda\mathbf{I})^{-1} = \mathbf{B}(\mathbf{B}^T\mathbf{B}+\lambda\mathbf{I})^{-1}\mathbf{B}^T = \mathbf{U}\boldsymbol{\Sigma}^2\left(\boldsymbol{\Sigma}^2 + \lambda\mathbf{I}\right)^{-1}\mathbf{U}^T$.

It is not hard to check (see [CLM+15]) that the ridge scores can be defined alternatively as:

$$l_i^\lambda = \min_{\mathbf{y}\in\mathbb{R}^n} \frac{1}{\lambda}\|\mathbf{b}_i^T - \mathbf{y}^T\mathbf{B}\|_2^2 + \|\mathbf{y}\|_2^2. \tag{4}$$

This formulation provides better insight into these scores. Since $\mathbf{B}\mathbf{B}^T = \mathbf{K}$, any kernel algorithm effectively works with $\mathbf{B}$'s rows as data points. The ridge scores reflect the relative importance of these rows. From (4) it's clear that $l_i^\lambda \leq 1$ since we can set $\mathbf{y}$ to the $i^{th}$ standard basis vector. $\mathbf{b}_i$ will have score $\ll 1$ (i.e. is less important) when it's possible to find a more "spread out" $\mathbf{y}$ that uses other rows in $\mathbf{B}$ to approximately reconstruct $\mathbf{b}_i$ – in other words when the row is less unique.

### 3.2 Sum of ridge leverage scores

As is standard in leverage score methods, we don't directly select landmarks to be the points with the highest scores. Instead, we sample each point with probability proportional to $l_i^\lambda$. Accordingly, the number of landmarks selected, which controls $\tilde{\mathbf{K}}$'s rank, is a random variable with expectation equal to the *sum of the $\lambda$-ridge leverage scores*. To ensure compact kernel approximations, we want this sum to be small. Immediately from Definition 1, we have:

**Fact 2.** $\qquad \sum_{i=1}^n l_i^\lambda(\mathbf{K}) = \text{tr}(\mathbf{K}(\mathbf{K} + \lambda\mathbf{I})^{-1}).$

We denote $d_{\text{eff}}^\lambda \stackrel{\text{def}}{=} \text{tr}(\mathbf{K}(\mathbf{K}+\lambda\mathbf{I})^{-1})$. $d_{\text{eff}}^\lambda$ is a natural quantity, referred to as the "effective dimension" or "degrees of freedom" for a ridge regression problem on $\mathbf{K}$ with regularization $\lambda$ [HTF02, Zha06]. $d_{\text{eff}}^\lambda$ increases monotonically as $\lambda$ decreases. For any fixed $\lambda$ it is essentially the smallest possible rank achievable for $\tilde{\mathbf{K}}$ satisfying the approximation guarantee given by RLS-Nyström: $\|\mathbf{K} - \tilde{\mathbf{K}}\|_2 < \lambda$.

## 3.3 The basic sampling algorithm

We can now introduce the RLS-Nyström method as Algorithm 1. We allow sampling each point by *any probability greater than $l_i^\lambda$*, which is useful later when we compute the scores approximately. Oversampling landmarks can only improve $\tilde{\mathbf{K}}$'s accuracy. It could cause us to take more samples, but we will always ensure that the sum of our approximate ridge leverage scores is not too large.

---

**Algorithm 1** RLS-NYSTRÖM SAMPLING

---

**input**: $\mathbf{x}_1, \ldots, \mathbf{x}_n \in \mathcal{X}$, kernel matrix $\mathbf{K}$, ridge parameter $\lambda > 0$, failure probability $\delta \in (0, 1/8)$

1: Compute an over-approximation, $\tilde{l}_i^\lambda > l_i^\lambda$ for the $\lambda$-ridge leverage score of each $\mathbf{x}_1, \ldots, \mathbf{x}_n$
2: Set $p_i := \min\left\{1, \tilde{l}_i^\lambda \cdot 16\log(\sum \tilde{l}_i^\lambda/\delta)\right\}$.
3: Construct $\mathbf{S} \in \mathbb{R}^{n \times s}$ by sampling $\mathbf{x}_1, \ldots, \mathbf{x}_n$ each independently with probability $p_i$. In other words, for each $i$ add a column to $\mathbf{S}$ with a 1 in position $i$ with probability $p_i$.
4: **return** the Nyström factors $\mathbf{KS} \in \mathbb{R}^{n \times s}$ and $(\mathbf{S}^T\mathbf{KS})^+ \in \mathbb{R}^{s \times s}$.

---

## 3.4 Accuracy bounds

We show that RLS-Nyström produces $\tilde{\mathbf{K}}$ which spectrally approximates $\mathbf{K}$ up to a small additive error. This is the strongest type of approximation offered by any known Nyström method [GM13]. It guarantees provable accuracy when $\tilde{\mathbf{K}}$ is used in place of $\mathbf{K}$ in many learning applications [CMT10].

**Theorem 3** (Spectral error approximation)**.** *For any $\lambda > 0$ and $\delta \in (0, 1/8)$, Algorithm 1 returns $\mathbf{S} \in \mathbb{R}^{n \times s}$ such that with probability $1 - \delta$, $s \leq 2\sum_i p_i$ and $\tilde{\mathbf{K}} = \mathbf{KS}(\mathbf{S}^T\mathbf{KS})^+\mathbf{S}^T\mathbf{K}$ satisfies:*

$$\tilde{\mathbf{K}} \preceq \mathbf{K} \preceq \tilde{\mathbf{K}} + \lambda\mathbf{I}. \tag{5}$$

*When ridge scores are computed exactly, $\sum_i p_i = O\left(d_{\text{eff}}^\lambda \log(d_{\text{eff}}^\lambda/\delta)\right)$.*

$\preceq$ denotes the Loewner ordering: $\mathbf{M} \preceq \mathbf{N}$ means that $\mathbf{N} - \mathbf{M}$ is positive semidefinite. Note that (5) immediately implies the well studied (see e.g [GM13]) spectral norm guarantee, $\|\mathbf{K} - \tilde{\mathbf{K}}\|_2 \leq \lambda$.

Intuitively, Theorem 3 guarantees that $\tilde{\mathbf{K}}$ well approximates the top of $\mathbf{K}$'s spectrum (i.e. any eigenvalues $> \lambda$) while losing information about smaller, less important eigenvalues. Due to space constraints, we defer the proof to Appendix A. It relies on the view of Nyström approximation as a low-rank projection of the kernelized data (see Section 2.1) and we use an intrinsic dimension matrix Bernstein bound to show accuracy of the sampled approximation.

Often the regularization parameter $\lambda$ is specified for a learning task, and for near optimal performance on this task, we set the approximation factor in Theorem 3 to $\epsilon\lambda$. In this case we have:

**Corollary 4** (Tighter spectral error approximation)**.** *For any $\lambda > 0$ and $\delta \in (0, 1/8)$, Algorithm 1 run with ridge parameter $\epsilon\lambda$ returns $\mathbf{S} \in \mathbb{R}^{n \times s}$ such that with probability $1 - \delta$, $s = O\left(\frac{d_{\text{eff}}^\lambda}{\epsilon} \log \frac{d_{\text{eff}}^\lambda}{\delta\epsilon}\right)$ and $\tilde{\mathbf{K}} = \mathbf{KS}(\mathbf{S}^T\mathbf{KS})^+\mathbf{S}^T\mathbf{K}$ satisfies $\tilde{\mathbf{K}} \preceq \mathbf{K} \preceq \tilde{\mathbf{K}} + \epsilon\lambda\mathbf{I}$.*

*Proof.* This follows from Theorem 3 by noting $d_{\text{eff}}^{\epsilon\lambda} \leq d_{\text{eff}}^\lambda/\epsilon$ since $(\mathbf{K}+\epsilon\lambda I)^{-1} \preceq \frac{1}{\epsilon}(\mathbf{K}+\lambda I)^{-1}$. □

Corollary 4 suffices to prove that $\tilde{\mathbf{K}}$ can be used in place of $\mathbf{K}$ without sacrificing performance on kernel ridge regression and canonical correlation tasks [AM15, Wan16]. We also use it to prove a *projection-cost preservation* guarantee (Theorem 12, Appendix B), which gives approximation bounds for kernel PCA and $k$-means clustering. Projection-cost preservation has proven a powerful concept in the matrix sketching literature [FSS13, CEM+15, CMM17, BWZ16, CW17] and we hope that extending the guarantee to kernels leads to applications beyond those considered in this work.

Our results on downstream learning bounds that can be derived from Theorem 3 are summarized in Table 1. Details can be found in Appendices B and E.

Table 1: Downstream guarantees for $\tilde{\mathbf{K}}$ obtained from RLS-Nyström (Algorithm 1).

| Application | Guarantee | Theorem | Space to store $\tilde{\mathbf{K}}$ |
|---|---|---|---|
| Kernel Ridge Regression w/ param $\lambda$ | $(1+\epsilon)$ relative error risk bound | Thm 16 | $\tilde{O}(\frac{n d_{\text{eff}}^{\lambda}}{\epsilon})$ |
| Kernel $k$-means Clustering | $(1+\epsilon)$ relative error | Thm 17 | $\tilde{O}(\frac{nk}{\epsilon})$ |
| Rank $k$ Kernel PCA | $(1+\epsilon)$ relative Frob norm error | Thm 18 | $\tilde{O}(\frac{nk}{\epsilon})$ |
| Kernel CCA w/ params $\lambda_x$, $\lambda_y$ | $\epsilon$ additive error | Thm 19 | $\tilde{O}(\frac{n d_{\text{eff}}^{\lambda_x} + n d_{\text{eff}}^{\lambda_y}}{\epsilon})$ |

$^*$ For conciseness, $\tilde{O}(\cdot)$ hides log factors in the failure probability, $d_{\text{eff}}$, and $k$.

## 4   Recursive sampling for efficient RLS-Nyström

Having established strong approximation guarantees for RLS-Nyström, it remains to provide an efficient implementation. Specifically, Step 1 of Algorithm 1 naively requires $\Theta(n^3)$ time. We show that significant acceleration is possible using a recursive sampling approach.

### 4.1   Ridge leverage score approximation via uniform sampling

The key is to estimate the leverage scores by computing (3) approximately, using a *uniform sample of the data points*. To ensure accuracy, the sample must be large – a constant fraction of the points. Our fast runtimes are achieved by recursively approximating this large sample. In Appendix F we prove:

**Lemma 5.** *For any* $\mathbf{B} \in \mathbb{R}^{n \times n}$ *with* $\mathbf{B}\mathbf{B}^T = \mathbf{K}$ *and* $\mathbf{S} \in \mathbb{R}^{n \times s}$ *chosen by sampling each data point independently with probability* $1/2$, *let* $\tilde{l}_i^\lambda = \mathbf{b}_i^T(\mathbf{B}^T\mathbf{S}\mathbf{S}^T\mathbf{B} + \lambda\mathbf{I})^{-1}\mathbf{b}_i$ *and* $p_i = \min\{1, 16\tilde{l}_i^\lambda \log(\sum_i \tilde{l}_i^\lambda/\delta)\}$ *for any* $\delta \in (0, 1/8)$. *Then with probability at least* $1 - \delta$:

$$1)\quad \tilde{l}_i^\lambda \geq l_i^\lambda \text{ for all } i \qquad 2)\quad \sum_i p_i \leq 64\sum_i l_i^\lambda \log(\sum_i l_i^\lambda/\delta).$$

The first condition ensures that the approximate scores $\tilde{l}_i^\lambda$ suffice for use in Algorithm 1. The second ensures that the Nyström approximation obtained will not have too many sampled landmarks.

Naively computing $\tilde{l}_i^\lambda$ in Lemma 5 involves explicitly forming $\mathbf{B}$, requiring $\Omega(n^2)$ time (e.g. $\Theta(n^3)$ via Cholesky decomposition). Fortunately, the following formula (proof in Appx. F) avoids this cost:

**Lemma 6.** *For any sampling matrix* $\mathbf{S} \in \mathbb{R}^{n \times s}$, *and any* $\lambda > 0$:

$$\tilde{l}_i^\lambda \overset{\text{def}}{=} \mathbf{b}_i^T(\mathbf{B}^T\mathbf{S}\mathbf{S}^T\mathbf{B} + \lambda\mathbf{I})^{-1}\mathbf{b}_i = \frac{1}{\lambda}\left(\mathbf{K} - \mathbf{KS}\left(\mathbf{S}^T\mathbf{KS} + \lambda\mathbf{I}\right)^{-1}\mathbf{S}^T\mathbf{K}\right)_{i,i}.$$

*It follows that we can compute all* $\tilde{l}_i^\lambda$ *for all* $i$ *in* $O(ns^2)$ *time using just* $O(ns)$ *kernel evaluations, to compute* $\mathbf{KS}$ *and the diagonal of* $\mathbf{K}$.

### 4.2   Recursive RLS-Nyström

We apply Lemmas 5 and 6 to give an efficient recursive implementation of RLS-Nyström, Algorithm 2. We show that the output of this algorithm, $\mathbf{S}$, is sampled according to approximate ridge leverage scores for $\mathbf{K}$ and thus satisfies the approximation guarantee of Theorem 3.

**Theorem 7** (Main Result). *Let* $\mathbf{S} \in \mathbb{R}^{n \times s}$ *be computed by Algorithm 2. With probability* $1 - 3\delta$, $s \leq 384 \cdot d_{\text{eff}}^\lambda \log(d_{\text{eff}}^\lambda/\delta)$, $\mathbf{S}$ *is sampled by overestimates of the* $\lambda$-*ridge leverage scores of* $\mathbf{K}$, *and thus by Theorem 3, the Nyström approximation* $\tilde{\mathbf{K}} = \mathbf{KS}(\mathbf{S}^T\mathbf{KS})^+\mathbf{S}^T\mathbf{K}$ *satisfies:*

$$\tilde{\mathbf{K}} \preceq \mathbf{K} \preceq \tilde{\mathbf{K}} + \lambda\mathbf{I}.$$

*Algorithm 2 uses* $O(ns)$ *kernel evaluations and* $O(ns^2)$ *computation time.*

**Algorithm 2** RECURSIVERLS-NYSTRÖM.

---

**input**: $\mathbf{x}_1, \ldots, \mathbf{x}_m \in \mathcal{X}$, kernel function $K : \mathcal{X} \times \mathcal{X} \to \mathbb{R}$, ridge $\lambda > 0$, failure prob. $\delta \in (0, 1/32)$
**output**: weighted sampling matrix $\mathbf{S} \in \mathbb{R}^{m \times s}$

1: **if** $m \leq 192 \log(1/\delta)$ **then**
2:      **return** $\mathbf{S} := \mathbf{I}_{m \times m}$.
3: **end if**
4: Let $\bar{\mathcal{S}}$ be a random subset of $\{1, ..., m\}$, with each $i$ included independently with probability $\frac{1}{2}$.
    ▷ Let $\bar{\mathbf{X}} = \{\mathbf{x}_{i_1}, \mathbf{x}_{i_2}, ..., \mathbf{x}_{i_{|\bar{\mathcal{S}}|}}\}$ for $i_j \in \bar{\mathcal{S}}$ be the data sample corresponding to $\bar{\mathcal{S}}$.
    ▷ Let $\bar{\mathbf{S}} = [\mathbf{e}_{i_1}, \mathbf{e}_{i_2}, ..., \mathbf{e}_{i_{|\bar{\mathcal{S}}|}}]$ be the sampling matrix corresponding to $\bar{\mathcal{S}}$.
5: $\tilde{\mathbf{S}} := $ RECURSIVERLS-NYSTRÖM$(\bar{\mathbf{X}}, K, \lambda, \delta/3)$.
6: $\hat{\mathbf{S}} := \bar{\mathbf{S}} \cdot \tilde{\mathbf{S}}$.
7: Set $\tilde{l}_i^\lambda := \frac{3}{2\lambda} \left( \mathbf{K} - \mathbf{K}\hat{\mathbf{S}} \left( \hat{\mathbf{S}}^T \mathbf{K}\hat{\mathbf{S}} + \lambda\mathbf{I} \right)^{-1} \hat{\mathbf{S}}^T \mathbf{K} \right)_{i,i}$ for each $i \in \{1, \ldots, m\}$.

    ▷ By Lemma 6, equals $\frac{3}{2}(\mathbf{B}(\mathbf{B}^T\hat{\mathbf{S}}\hat{\mathbf{S}}^T\mathbf{B} + \lambda\mathbf{I})^{-1}\mathbf{B}^T)_{i,i}$. $\mathbf{K}$ denotes the kernel matrix for data-points $\{\mathbf{x}_1, \ldots, \mathbf{x}_m\}$ and kernel function $K$.
8: Set $p_i := \min\{1, \tilde{l}_i^\lambda \cdot 16 \log(\sum \tilde{l}_i^\lambda / \delta)\}$ for each $i \in \{1, \ldots, m\}$.
9: Initially set weighted sampling matrix $\mathbf{S}$ to be empty. For each $i \in \{1, \ldots, m\}$, with probability $p_i$, append the column $\frac{1}{\sqrt{p_i}}\mathbf{e}_i$ onto $\mathbf{S}$.
10: **return** $\mathbf{S}$.

---

Note that in Algorithm 2 the columns of $\mathbf{S}$ are weighted by $1/\sqrt{p_i}$. The Nyström approximation $\tilde{\mathbf{K}} = \mathbf{K}\mathbf{S}(\mathbf{S}^T\mathbf{K}\mathbf{S})^+\mathbf{S}^T\mathbf{K}$ is not effected by column weights (see derivation in Section 2.1). However, the weighting is necessary when the output is used in recursive calls (i.e. when $\tilde{\mathbf{S}}$ is used in Step 6).

We prove Theorem 7 via the following intermediate result:

**Theorem 8.** *For any inputs* $\mathbf{x}_1, \ldots, \mathbf{x}_m$, $K$, $\lambda > 0$ *and* $\delta \in (0, 1/32)$, *let* $\mathbf{K}$ *be the kernel matrix for* $\mathbf{x}_1, \ldots, \mathbf{x}_m$ *and kernel function* $K$ *and let* $d_{\text{eff}}^\lambda(\mathbf{K})$ *be the effective dimension of* $\mathbf{K}$ *with parameter* $\lambda$. *With probability* $(1 - 3\delta)$, RECURSIVERLS-NYSTRÖM *outputs* $\mathbf{S}$ *with* $s$ *columns that satisfies:*

$$\frac{1}{2}(\mathbf{B}^T\mathbf{B} + \lambda\mathbf{I}) \preceq (\mathbf{B}^T\mathbf{S}\mathbf{S}^T\mathbf{B} + \lambda\mathbf{I}) \preceq \frac{3}{2}(\mathbf{B}^T\mathbf{B} + \lambda\mathbf{I}) \qquad \text{for any } \mathbf{B} \text{ with } \mathbf{B}\mathbf{B}^T = \mathbf{K}. \quad (6)$$

*Additionally,* $s \leq s_{\max}(d_{\text{eff}}^\lambda(\mathbf{K}), \delta)$ *where* $s_{\max}(w, z) \stackrel{\text{def}}{=} 384 \cdot (w + 1) \log((w + 1)/z)$. *The algorithm uses* $\leq c_1 m s_{\max}(d_{\text{eff}}^\lambda(\mathbf{K}), \delta)$ *kernel evaluations and* $\leq c_2 m s_{\max}(d_{\text{eff}}^\lambda(\mathbf{K}), \delta)^2$ *additional computation time where* $c_1$ *and* $c_2$ *are fixed universal constants.*

Theorem 8 is proved via an inductive argument, given in Appendix C. Roughly, consider in Step 6 of Algorithm 2, setting $\hat{\mathbf{S}} := \bar{\mathbf{S}}$ instead of $\bar{\mathbf{S}} \cdot \tilde{\mathbf{S}}$. By Lemma 5 and the formula in Lemma 6, the leverage score approximations $\tilde{l}_i^\lambda$ computed in Step 7 would be good approximations to the true leverage scores, and $\mathbf{S}$ would satisfy Theorem 8 by a standard matrix Bernstein bound (see Lemma 9).

However, if we set $\hat{\mathbf{S}} := \bar{\mathbf{S}}$, it will have $n/2$ columns in expectation, and the computation in Step 7 will be expensive – requiring roughly $O(n^3)$ time. By recursively calling Algorithm 8 and applying Theorem 8 inductively, we obtain $\tilde{\mathbf{S}}$ satisfying with high probability:

$$\frac{1}{2}(\mathbf{B}^T\bar{\mathbf{S}}\bar{\mathbf{S}}^T\mathbf{B} + \lambda\mathbf{I}) \preceq ((\mathbf{B}\bar{\mathbf{S}})\tilde{\mathbf{S}}\tilde{\mathbf{S}}^T(\bar{\mathbf{S}}^T\mathbf{B}) + \lambda\mathbf{I}) \preceq \frac{3}{2}(\mathbf{B}\bar{\mathbf{S}}\bar{\mathbf{S}}^T\mathbf{B} + \lambda\mathbf{I}).$$

This guarantee ensures that when we use $\hat{\mathbf{S}} = \hat{\mathbf{S}} \cdot \tilde{\mathbf{S}}$ in place of $\bar{\mathbf{S}}$ in Step 7, the leverage score estimates are changed only by a constant factor. Thus, sampling by these estimates, still gives us the desired guarantee (6). Further, $\tilde{\mathbf{S}}$ and therefore $\hat{\mathbf{S}}$ has just $O(s_{\max}(d_{\text{eff}}^\lambda(\mathbf{K}), \delta))$ columns, so Step 7 can be performed very efficiently, within the stated runtime bounds.

With Theorem 8 we can easily prove our main result, Theorem 7.

*Proof of Theorem 7.* In our proof of Theorem 3 in Appendix A.1, we show that if

$$\frac{1}{2}(\mathbf{B}^T\mathbf{B} + \lambda\mathbf{I}) \preceq (\mathbf{B}^T\mathbf{S}\mathbf{S}^T\mathbf{B} + \lambda\mathbf{I}) \preceq \frac{3}{2}(\mathbf{B}^T\mathbf{B} + \lambda\mathbf{I})$$

for a weighted sampling matrix $\mathbf{S}$, then even if we remove the weights from $\mathbf{S}$ so that it has all unit entries (they don't effect the Nyström approximation), $\tilde{\mathbf{K}} = \mathbf{KS}(\mathbf{S}^T\mathbf{KS})^+\mathbf{S}^T\mathbf{K}$ satisfies:

$$\tilde{\mathbf{K}} \preceq \mathbf{K} \preceq \tilde{\mathbf{K}} + \lambda\mathbf{I}.$$

The runtime bounds also follow nearly directly from Theorem 8. In particular, we have established that $O\left(ns_{\max}(d_{\text{eff}}^\lambda(\mathbf{K}),\delta)\right)$ kernel evaluations and $O\left(ns_{\max}(d_{\text{eff}}^\lambda(\mathbf{K}),\delta)^2\right)$ additional runtime are required by RECURSIVERLS-NYSTRÖM. We only needed the upper bound to prove Theorem 8, but along the way actually show that in a successful run of RECURSIVERLS-NYSTRÖM, $\mathbf{S}$ has $\Theta\left(d_{\text{eff}}^\lambda(\mathbf{K})\log\left(d_{\text{eff}}^\lambda(\mathbf{K})/\delta\right)\right)$ columns. Additionally, we may assume that $d_{\text{eff}}(\mathbf{K}) \geq 1/2$. If it is not, then it's not hard to check (see proof of Lemma 20) that $\lambda$ must be $\geq \|\mathbf{K}\|$. If this is the case, the guarantee of Theorem 7 is vacuous: *any* Nyström approximation $\tilde{\mathbf{K}}$ satisfies $\tilde{\mathbf{K}} \preceq \mathbf{K} \preceq \tilde{\mathbf{K}} + \lambda\mathbf{I}$. With $d_{\text{eff}}(\mathbf{K}) \geq 1/2$, $d_{\text{eff}}^\lambda(\mathbf{K})\log\left(d_{\text{eff}}^\lambda(\mathbf{K})/\delta\right)$ and thus $s$ are $\Theta(s_{\max}(d_{\text{eff}}^\lambda(\mathbf{K}),\delta)$ so we conclude that Theorem 7 uses $O(ns)$ kernel evaluations and $O(ns^2)$ additional runtime. $\qquad\square$

## 5 Empirical evaluation

We conclude with an empirical evaluation of our recursive RLS-Nyström method. We use a variant of Algorithm 2 where, instead of choosing a regularization parameter $\lambda$, the user sets a sample size $s$ and $\lambda$ is automatically determined such that $s = \Theta(d_{\text{eff}}^\lambda \cdot \log(d_{\text{eff}}^\lambda/\delta))$. This variant is practically appealing as it essentially yields the best possible approximation to $\mathbf{K}$ for a fixed sample budget. Pseudocode and proofs of correctness are included in Appendix D.

### 5.1 Performance of Recursive RLS-Nyström for kernel approximation

We evaluate RLS-Nyström on the `YearPredictionMSD`, `Covertype`, `Cod-RNA`, and `Adult` datasets downloaded from the UCI ML Repository [Lic13] and [UKM06]. These datasets contain $515345$, $581012$, $331152$, and $48842$ data points respectively. We compare against the classic Nyström method with uniform sampling [WS01] and the random Fourier features method [RR07]. Due to the large size of the datasets, prior leverage score based Nyström approaches [DMIMW12, GM13, AM15], which require at least $\Omega(n^2)$ time are infeasible, and thus not included in our tests.

We split categorical features into binary indicatory features and mean center and normalize features to have variance 1. We use a Gaussian kernel for all tests, with the width parameter $\sigma$ selected via cross validation on regression and classification tasks. $\|\mathbf{K} - \tilde{\mathbf{K}}\|_2$ is used to measure approximation error. Since this quantity is prohibitively expensive to compute directly (it requires building the full kernel matrix $\mathbf{K}$), the error is estimated using a random subset of 20,000 data points and repeated trials.

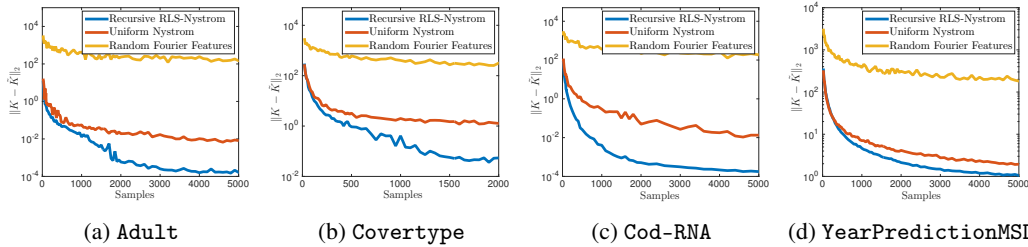

(a) `Adult`    (b) `Covertype`    (c) `Cod-RNA`    (d) `YearPredictionMSD`

Figure 2: For a given number of samples, Recursive RLS-Nyström yields approximations with lower error, measured by $\|\mathbf{K} - \tilde{\mathbf{K}}\|_2$. Error is plotted on a logarithmic scale, averaged over 10 trials.

Figure 2 confirms that Recursive RLS-Nyström consistently obtains substantially better kernel approximation error than the other methods. As we can see in Figure 3, with the exception of `YearPredictionMSD`, the better quality of the landmarks obtained with Recursive RLS-Nyström also translates into runtime improvements. While the cost *per sample* is higher for our method at $O(nd + ns)$ time versus $O(nd + s^2)$ for uniform Nyström and $O(nd)$ for random Fourier features, since RLS-Nyström requires fewer samples it more quickly obtains $\tilde{\mathbf{K}}$ with a given accuracy. $\tilde{\mathbf{K}}$ will also have lower rank, which can accelerate processing in downstream applications.

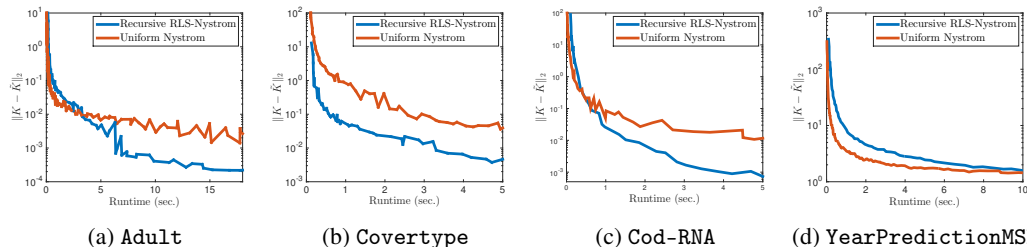

(a) `Adult`    (b) `Covertype`    (c) `Cod-RNA`    (d) `YearPredictionMSD`

Figure 3: Recursive RLS-Nyström obtains a fixed level of approximation faster than uniform sampling, only underperforming on `YearPredictionMSD`. Results for random Fourier features are not shown: while the method is faster, it never obtained high enough accuracy to be directly comparable.

In Appendix G, we show that that runtime of RLS-Nyström can be further accelerated, via a heuristic approach that under-samples landmarks at each level of recursion. This approach brings the per sample cost down to approximately that of random Fourier features and uniform Nyström while nearly maintaining the same approximation quality. Results are shown in Figure 4.

For datasets such as `Covertype` in which Recursive RLS-Nyström performs significantly better than uniform sampling, so does the accelerated method (see Figure 4b). However, the performance of the accelerated method does not degrade when leverage scores are relatively uniform – it still offers the best runtime to approximation quality tradeoff (Figure 4c).

We note further runtime optimizations may be possible. Subsequent work extends fast ridge leverage score methods to distributed and streaming environments [CLV17]. Empirical evaluation of these techniques could lead to even more scalable, high accuracy Nyström methods.

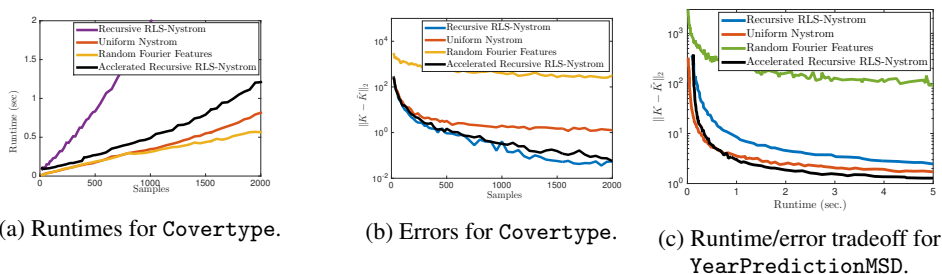

(a) Runtimes for `Covertype`.    (b) Errors for `Covertype`.    (c) Runtime/error tradeoff for `YearPredictionMSD`.

Figure 4: Our accelerated Recursive RLS-Nyström, nearly matches the *per sample runtime* of random Fourier features and uniform Nyström while still providing much better approximation.

## 5.2 Additional Empirical Results

In Appendix G we verify the usefulness of our kernel approximations in downstream learning tasks. While full kernel methods do not scale to our large datasets, Recursive RLS-Nyström does since its runtime depends linearly on $n$. For example, on `YearPredictionMSD` the method requires 307 sec. (averaged over 5 trials) to build a 2,000 landmark Nyström approximation for 463,716 training points. Ridge regression using the approximate kernel then requires 208 sec. for a total of 515 sec. These runtime are comparable to those of the very fast random Fourier features method, which underperforms RLS-Nyström in terms of regression and classification accuracy.

## Acknowledgements

We would like to thank Michael Mahoney for bringing the potential of ridge leverage scores to our attention and suggesting their possible approximation via iterative sampling schemes. We would also like to thank Michael Cohen for pointing out (and fixing) an error in our original manuscript and generally for his close collaboration in our work on leverage score sampling algorithms. Finally, thanks to Haim Avron for pointing our an error in our original analysis.

## Footnotes

[1]This is within a log factor of the best possible for any low-rank approximation with error $\lambda$.

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
