[Supplementary Material]

# A Ridge leverage score sampling bounds

Here we give the primary matrix concentration results used to bound the performance of ridge leverage score sampling in Theorems 3, 7, and 14.

**Lemma 9.** *For any $\lambda > 0$ and $\delta \in (0, 1/8)$, given ridge leverage score approximations $\tilde{l}_i^\lambda \geq \ell_i^\lambda$ for all $i$, let $p_i = \min\left\{1, 16\tilde{l}_i^\lambda \log(\sum \tilde{l}_i^\lambda / \delta)\right\}$. Let $\mathbf{S} \in \mathbb{R}^{n \times s}$ be selected by sampling the standard basis vectors $\mathbf{e}_1, \ldots, \mathbf{e}_n$ each independently with probability $p_i$ and rescaling selected columns by $1/\sqrt{p_i}$. With probability $1 - \delta$, $1/2 \cdot \sum_i p_i \leq s \leq 2 \sum_i p_i$ and:*

$$\frac{1}{2}\mathbf{B}^T\mathbf{B} - \frac{1}{2}\lambda\mathbf{I} \preceq \mathbf{B}^T\mathbf{S}\mathbf{S}^T\mathbf{B} \preceq \frac{3}{2}\mathbf{B}^T\mathbf{B} + \frac{1}{2}\lambda\mathbf{I}, \qquad (7)$$

*Proof.* Let $\mathbf{B} = \mathbf{U}\boldsymbol{\Sigma}\mathbf{V}^T$ be the singular value decomposition of $\mathbf{B}$. By Definition 1:

$$l_i^\lambda = \mathbf{b}_i^T \left(\mathbf{B}^T\mathbf{B} + \lambda\mathbf{I}\right)^{-1} \mathbf{b}_i = \mathbf{b}_i^T \left(\mathbf{V}\boldsymbol{\Sigma}^2\mathbf{V}^T + \lambda\mathbf{V}\mathbf{V}^T\right)^{-1} \mathbf{b}_i$$
$$= \mathbf{b}_i^T \left(\mathbf{V}\bar{\boldsymbol{\Sigma}}^2\mathbf{V}^T\right)^{-1} \mathbf{b}_i$$
$$= \mathbf{b}_i^T \left(\mathbf{V}\bar{\boldsymbol{\Sigma}}^{-2}\mathbf{V}^T\right) \mathbf{b}_i,$$

where $\bar{\boldsymbol{\Sigma}}_{i,i}^2 = \sigma_i^2(\mathbf{B}) + \lambda$. For each $i \in 1, \ldots, n$ define the matrix valued random variable:

$$\mathbf{X}_i = \begin{cases} \left(\frac{1}{p_i} - 1\right) \bar{\boldsymbol{\Sigma}}^{-1}\mathbf{V}^T\mathbf{b}_i\mathbf{b}_i^T\mathbf{V}\bar{\boldsymbol{\Sigma}}^{-1} & \text{with probability } p_i \\ -\bar{\boldsymbol{\Sigma}}^{-1}\mathbf{V}^T\mathbf{b}_i\mathbf{b}_i^T\mathbf{V}\bar{\boldsymbol{\Sigma}}^{-1} & \text{with probability } (1 - p_i) \end{cases}$$

Let $\mathbf{Y} = \sum_i \mathbf{X}_i$. We have $\mathbb{E}\,\mathbf{Y} = \mathbf{0}$. Furthermore, $\mathbf{B}^T\mathbf{S}\mathbf{S}^T\mathbf{B} = \mathbf{V}\bar{\boldsymbol{\Sigma}}\mathbf{Y}\bar{\boldsymbol{\Sigma}}\mathbf{V}^T + \mathbf{B}^T\mathbf{B}$. If we can show that $\|\mathbf{Y}\|_2 \leq \frac{1}{2}$, then since $\mathbf{V}\bar{\boldsymbol{\Sigma}}^2\mathbf{V}^T = \mathbf{B}^T\mathbf{B} + \lambda\mathbf{I}$ this would give the desired bound:

$$\frac{1}{2}\mathbf{B}^T\mathbf{B} - \frac{1}{2}\lambda\mathbf{I} \preceq \mathbf{B}^T\mathbf{S}\mathbf{S}^T\mathbf{B} \preceq \frac{3}{2}\mathbf{B}^T\mathbf{B} + \frac{1}{2}\lambda\mathbf{I}.$$

To prove that $\|\mathbf{Y}\|_2$ is small we use an intrinsic dimension matrix Bernstein inequality. This inequality will bound the deviation of $\mathbf{Y}$ from its expectation as long as we can bound each $\|\mathbf{X}_i\|_2$ and we can bound the matrix variance $\mathbb{E}(\mathbf{Y}^2)$.

**Theorem 10** (Theorem 7.3.1, [Tro15]). *Let $\mathbf{X}_1, \ldots, \mathbf{X}_n$ be random symmetric matrices such that for all $i$, $\mathbb{E}\,\mathbf{X} = \mathbf{0}$ and $\|\mathbf{X}_i\|_2 \leq L$. Let $\mathbf{Y} = \sum_{i=1}^n \mathbf{X}_i$. As long we can bound the matrix variance:*

$$\mathbb{E}(\mathbf{Y}^2) \preceq \mathbf{Z},$$

*then for for $t \geq \sqrt{\|\mathbf{Z}\|_2} + L/3$,*

$$\mathbb{P}\left[\|\mathbf{Y}\| \geq t\right] \leq 4\frac{\text{tr}(\mathbf{Z})}{\|\mathbf{Z}\|_2} e^{\frac{-t^2/2}{\|\mathbf{Z}\|_2 + Lt/3}}.$$

If $p_i = 1$ (i.e. $c\tilde{l}_i^\lambda \log(\sum \tilde{l}_i^\lambda / \delta) \geq 1$) then $\mathbf{X}_i = \mathbf{0}$ so $\|\mathbf{X}_i\|_2 = 0$. Otherwise, we use the fact that:

$$\frac{1}{\tilde{l}_i^\lambda}\mathbf{b}_i\mathbf{b}_i^T \preceq \frac{1}{l_i^\lambda}\mathbf{b}_i\mathbf{b}_i^T \preceq \mathbf{B}^T\mathbf{B} + \lambda\mathbf{I}. \qquad (8)$$

This follows because we can write any $\mathbf{x}$ as $\mathbf{x} = (\mathbf{B}^T\mathbf{B} + \lambda\mathbf{I})^{-1/2}\mathbf{y}$ for some $\mathbf{y}$. We can then write:

$$\mathbf{x}^T\mathbf{b}_i\mathbf{b}_i^T\mathbf{x} = \mathbf{y}^T(\mathbf{B}^T\mathbf{B} + \lambda\mathbf{I})^{-1/2}\mathbf{b}_i\mathbf{b}_i^T(\mathbf{B}^T\mathbf{B} + \lambda\mathbf{I})^{-1/2}\mathbf{y}$$
$$\leq \|\mathbf{y}\|_2^2 \cdot \|(\mathbf{B}^T\mathbf{B} + \lambda\mathbf{I})^{-1/2}\mathbf{b}_i\mathbf{b}_i^T(\mathbf{B}^T\mathbf{B} + \lambda\mathbf{I})^{-1/2}\|_2.$$

Since $(\mathbf{B}^T\mathbf{B} + \lambda\mathbf{I})^{-1/2}\mathbf{b}_i\mathbf{b}_i^T(\mathbf{B}^T\mathbf{B} + \lambda\mathbf{I})^{-1/2}$ is rank 1, we have:

$$\|(\mathbf{B}^T\mathbf{B} + \lambda\mathbf{I})^{-1/2}\mathbf{b}_i\mathbf{b}_i^T(\mathbf{B}^T\mathbf{B} + \lambda\mathbf{I})^{-1/2}\|_2 = \text{tr}\left((\mathbf{B}^T\mathbf{B} + \lambda\mathbf{I})^{-1/2}\mathbf{b}_i\mathbf{b}_i^T(\mathbf{B}^T\mathbf{B} + \lambda\mathbf{I})^{-1/2}\right)$$
$$= \mathbf{b}_i^T(\mathbf{B}^T\mathbf{B} + \lambda\mathbf{I})^{-1}\mathbf{b}_i = l_i^\lambda \qquad (9)$$

where in the last step we use the cyclic property of the trace. Writing $\mathbf{y} = (\mathbf{B}^T\mathbf{B} + \lambda\mathbf{I})^{1/2}\mathbf{x}$ and plugging back into (9) gives:

$$\mathbf{x}^T\mathbf{b}_i\mathbf{b}_i^T\mathbf{x} \leq \|\mathbf{y}\|_2^2 \cdot l_i^\lambda = \mathbf{x}^T(\mathbf{B}^T\mathbf{B} + \lambda\mathbf{I})\mathbf{x} \cdot l_i^\lambda.$$

Rearranging and using that $\tilde{l}_i^\lambda \geq l_i^\lambda$ gives (8). With this bound in place we get:

$$\frac{1}{\tilde{l}_i^\lambda} \cdot \bar{\mathbf{\Sigma}}^{-1}\mathbf{V}^T\mathbf{b}_i\mathbf{b}_i^T\mathbf{V}\bar{\mathbf{\Sigma}}^{-1} \preceq \bar{\mathbf{\Sigma}}^{-1}\mathbf{V}^T\left(\mathbf{B}^T\mathbf{B} + \lambda\mathbf{I}\right)\mathbf{V}\bar{\mathbf{\Sigma}}^{-1} = \mathbf{I}.$$

So we have:

$$\mathbf{X}_i \preceq \frac{1}{p_i}\bar{\mathbf{\Sigma}}^{-1}\mathbf{V}^T\mathbf{b}_i\mathbf{b}_i^T\mathbf{V}\bar{\mathbf{\Sigma}}^{-1} \preceq \frac{\tilde{l}_i^\lambda}{p_i}\mathbf{I} = \frac{1}{16\log\left(\sum \tilde{l}_i^\lambda/\delta\right)}\mathbf{I} \preceq \frac{1}{16\log\left(\sum l_i^\lambda/\delta\right)}\mathbf{I}.$$

Next we bound the variance of $\mathbf{Y}$.

$$\mathbb{E}(\mathbf{Y}^2) = \sum \mathbb{E}(\mathbf{X}_i^2) \preceq \sum \left[p_i\left(\frac{1}{p_i} - 1\right)^2 + (1 - p_i)\right] \cdot \bar{\mathbf{\Sigma}}^{-1}\mathbf{V}^T\mathbf{b}_i\mathbf{b}_i^T\mathbf{V}\bar{\mathbf{\Sigma}}^{-2}\mathbf{V}^T\mathbf{b}_i\mathbf{b}_i^T\mathbf{V}\bar{\mathbf{\Sigma}}^{-1}$$

$$\preceq \sum \frac{1}{p_i} \cdot l_i^{\tilde{\lambda}} \cdot \bar{\mathbf{\Sigma}}^{-1}\mathbf{V}^T\mathbf{b}_i\mathbf{b}_i^T\mathbf{V}\bar{\mathbf{\Sigma}}^{-1} \preceq \frac{1}{16\log\left(\sum l_i^\lambda/\delta\right)}\bar{\mathbf{\Sigma}}^{-1}\mathbf{V}^T\mathbf{B}^T\mathbf{B}\mathbf{V}\bar{\mathbf{\Sigma}}^{-1}$$

$$\preceq \frac{1}{16\log\left(\sum l_i^\lambda/\delta\right)}\mathbf{\Sigma}^2\bar{\mathbf{\Sigma}}^{-2} \preceq \frac{1}{16\log\left(\sum l_i^\lambda/\delta\right)}\mathbf{D}. \tag{10}$$

where $\mathbf{D}_{1,1} = 1$ and $\mathbf{D}_{i,i} = (\mathbf{\Sigma}^2\bar{\mathbf{\Sigma}}^{-2})_{i,i} = \frac{\sigma_i^2(\mathbf{B})}{\sigma_i^2(\mathbf{B})+\lambda}$ for all $i \geq 2$. Note that $\|\mathbf{D}\|_2 = 1$.

Then applying Theorem 10 with $\mathbf{Z} = \mathbf{D}/16\log\left(\sum l_i^\lambda/\delta\right)$ we see that:

$$\mathbb{P}\left[\|\mathbf{Y}\|_2 \geq \frac{1}{2}\right] \leq 4\,\mathrm{tr}(\mathbf{D})e^{\frac{-1/8}{\frac{1}{16\log(\sum l_i^\lambda/\delta)} + \frac{1}{192\log(\sum l_i^\lambda/\delta)}}}. \tag{11}$$

Then we observe that:

$$\mathrm{tr}(\mathbf{D}) \leq 1 + \mathrm{tr}(\mathbf{\Sigma}^2\bar{\mathbf{\Sigma}}^{-2}) = 1 + \mathrm{tr}\left(\mathbf{K}(\mathbf{K} + \lambda\mathbf{I})^{-1}\right) = 1 + \sum_i l_i^\lambda.$$

Plugging into (11), establishes (7):

$$\mathbb{P}\left[\|\mathbf{Y}\| \geq \frac{1}{2}\right] \leq 4\left(1 + \sum_i l_i^\lambda\right) \cdot e^{-2\log(\sum l_i^\lambda/\delta)} \leq \delta/2.$$

Note that here we make the extremely mild assumption that $\sum_i l_i^\lambda \geq 1$. If not, we can simply use a smaller $\lambda$ that makes this condition true, and will have $s = O(1)$.

All that remains to show is that the sample size $s$ is bounded with high probability. If $p_i = 1$, we always sample $i$ so there is no variance in $s$. Let $S \subseteq [1, ..., n]$ be the set of indices with $p_i < 1$. The expected number of points sampled from $S$ is $\sum_{i\in S} p_i = 16\log(\sum \tilde{l}_i^\lambda/\delta)\sum_{i\in S} \tilde{l}_i^\lambda$. Assume without loss of generality that $\sum_{i\in S} \tilde{l}_i^\lambda \geq 1$ – otherwise can just increase our leverage score estimates and increase the expected sample size by at most 1. Then, by a standard Chernoff bound, with probability at least $1 - \delta/2$,

$$\frac{1}{2} \cdot 16\log(\sum \tilde{l}_i^\lambda/\delta)\sum_{i\in S} \tilde{l}_i^\lambda \leq s \leq 2 \cdot 16\log(\sum \tilde{l}_i^\lambda/\delta)\sum_{i\in S} \tilde{l}_i^\lambda.$$

Union bounding over failure probabilities gives the lemma. $\qquad\square$

Lemma 9 yields an easy corollary about sampling *without rescaling* the columns in $\mathbf{S}$:

**Corollary 11.** *For any $\lambda > 0$ and $\delta \in (0, 1/8)$, given ridge leverage score approximations $\tilde{l}_i^\lambda \geq \frac{\lambda}{i}$ for all $i$, let $p_i = \min\left\{16\tilde{l}_i^\lambda\log(\sum \tilde{l}_i^\lambda/\delta), 1\right\}$. Let $\mathbf{S} \in \mathbb{R}^{n\times s}$ be selected by sampling, **but not rescaling**, the standard basis vectors $\mathbf{e}_1, \ldots, \mathbf{e}_n$ each independently with probability $p_i$. With probability $1 - \delta$, $1/2 \cdot \sum_i p_i \leq s \leq 2\sum_i p_i$ and there exists some scaling factor $C > 0$ such that*

$$\mathbf{B}^T\mathbf{B} \preceq C \cdot \mathbf{B}^T\mathbf{S}\mathbf{S}^T\mathbf{B} + \lambda\mathbf{I}. \tag{12}$$

*Proof.* By Lemma 9, if we set $C' = \frac{1}{\min_i p_i}$ we have:

$$\frac{1}{2}\mathbf{B}^T\mathbf{B} - \frac{1}{2}\lambda\mathbf{I} \preceq C' \cdot \mathbf{B}^T\mathbf{SS}^T\mathbf{B}$$

$$\mathbf{B}^T\mathbf{B} \preceq 2C' \cdot \mathbf{B}^T\mathbf{SS}^T\mathbf{B} + \lambda\mathbf{I}$$

which gives the corollary by setting $C = 2C'$. $\qquad\square$

## A.1 Spectral Error Kernel Approximation

We now give the deferred proof of Theorem 3, our main approximation bound, using the matrix concentration results proven above.

**Theorem 3** (Spectral error approximation). *For any $\lambda > 0$ and $\delta \in (0, 1/8)$, Algorithm 1 returns $\mathbf{S} \in \mathbb{R}^{n \times s}$ such that with probability $1 - \delta$, $s \leq 2\sum_i p_i$ and $\tilde{\mathbf{K}} := \mathbf{KS}(\mathbf{S}^T\mathbf{KS})^+\mathbf{S}^T\mathbf{K}$ satisfies:*

$$\tilde{\mathbf{K}} \preceq \mathbf{K} \preceq \tilde{\mathbf{K}} + \lambda\mathbf{I}. \tag{13}$$

*When ridge scores are computed exactly, $\sum_i p_i = O\left(d_{eff}^\lambda \log(d_{eff}^\lambda/\delta)\right)$.*

*Proof.* It is clear from the view of Nyström approximation as a low-rank projection of the kernelized data (see Section 2.1) that $\tilde{\mathbf{K}} \preceq \mathbf{K}$. Formally, for any $\mathbf{B} \in \mathbb{R}^{n \times n}$ with $\mathbf{BB}^T = \mathbf{K}$:

$$\tilde{\mathbf{K}} = \mathbf{KS}(\mathbf{S}^T\mathbf{KS})^+\mathbf{S}^T\mathbf{K} = \mathbf{BP_S}\mathbf{B}^T,$$

where $\mathbf{P_S} = \mathbf{B}^T\mathbf{S}(\mathbf{S}^T\mathbf{BB}^T\mathbf{S})^+\mathbf{S}^T\mathbf{B}$ is the orthogonal projection onto the row span of $\mathbf{S}^T\mathbf{B}$. Since $\mathbf{P_S}$ is a projection $\|\mathbf{P_S}\|_2 \leq 1$. So, for any $\mathbf{x} \in \mathbb{R}^n$:

$$\mathbf{x}^T\tilde{\mathbf{K}}\mathbf{x} = \mathbf{x}^T\mathbf{BP_S}\mathbf{Bx} = \|\mathbf{P_S}\mathbf{Bx}\|_2^2 \leq \|\mathbf{Bx}\|_2^2 = \mathbf{x}^T\mathbf{Kx},$$

which is equivalent to $\tilde{\mathbf{K}} \preceq \mathbf{K}$. It remains to show that $\mathbf{K} \preceq \tilde{\mathbf{K}} + \lambda\mathbf{I}$.

In Lemma 9 above, we apply a matrix Bernstein bound [Tro15] to prove that, when $\mathbf{S}$'s columns are reweighted by the inverse of their sampling probabilities, with probability $1 - \delta/2$:

$$\frac{1}{2}\left(\mathbf{B}^T\mathbf{B} + \lambda\mathbf{I}\right) \preceq \mathbf{B}^T\mathbf{SS}^T\mathbf{B} + \lambda\mathbf{I} \preceq \frac{3}{2}\left(\mathbf{B}^T\mathbf{B} + \lambda\mathbf{I}\right).$$

By Corollary 11, even if $\mathbf{S}$ is unweighted, as in Algorithm 1, this bound implies the existence of some finite scaling factor $C > 0$ such that:

$$\mathbf{B}^T\mathbf{B} \preceq C \cdot \mathbf{B}^T\mathbf{SS}^T\mathbf{B} + \lambda\mathbf{I}. \tag{14}$$

Let $\bar{\mathbf{P}}_\mathbf{S} = \mathbf{I} - \mathbf{P_S}$ be the projection onto the complement of the row span of $\mathbf{S}^T\mathbf{B}$. By (14):

$$\bar{\mathbf{P}}_\mathbf{S}\mathbf{B}^T\mathbf{B}\bar{\mathbf{P}}_\mathbf{S} \preceq C \cdot \bar{\mathbf{P}}_\mathbf{S}\mathbf{B}^T\mathbf{SS}^T\mathbf{B}\bar{\mathbf{P}}_\mathbf{S} + \lambda\bar{\mathbf{P}}_\mathbf{S}\mathbf{I}\bar{\mathbf{P}}_\mathbf{S}. \tag{15}$$

Since $\bar{\mathbf{P}}_\mathbf{S}$ projects to the complement of the row span of $\mathbf{S}^T\mathbf{B}$, $\mathbf{S}^T\mathbf{B}\bar{\mathbf{P}}_\mathbf{S} = \mathbf{0}$. So (15) gives:

$$\bar{\mathbf{P}}_\mathbf{S}\mathbf{B}^T\mathbf{B}\bar{\mathbf{P}}_\mathbf{S} \preceq \mathbf{0} + \lambda\bar{\mathbf{P}}_\mathbf{S}\mathbf{I}\bar{\mathbf{P}}_\mathbf{S} \preceq \lambda\mathbf{I}.$$

In other notation, $\|\bar{\mathbf{P}}_\mathbf{S}\mathbf{B}^T\mathbf{B}\bar{\mathbf{P}}_\mathbf{S}\|_2 \leq \lambda$. This in turn implies $\|\mathbf{B}\bar{\mathbf{P}}_\mathbf{S}\mathbf{B}^T\|_2 \leq \lambda$ and so $\mathbf{B}\bar{\mathbf{P}}_\mathbf{S}\mathbf{B}^T \preceq \lambda\mathbf{I}$. Rearranging and using $\mathbf{K} = \mathbf{BB}^T$ and $\tilde{\mathbf{K}} = \mathbf{BP_S}\mathbf{B}^T$ gives the result. $\qquad\square$

## B  Projection-cost preserving kernel approximation

In addition to the basic spectral approximation guarantee of Theorem 3, we prove that, with high probability, the RLS-Nyström method presented in Algorithm 1 outputs an approximation $\tilde{\mathbf{K}}$ satisfying what is known as a *projection-cost preservation guarantee*.

**Theorem 12** (Projection-cost preserving kernel approximation). *Let $\lambda = \frac{\epsilon}{k}\sum_{i=k+1}^n \sigma_i(\mathbf{K})$. For any $\epsilon \in (0, 1), \delta \in (0, 1/8)$, RLS-Nyström returns an $\mathbf{S} \in \mathbb{R}^{n \times s}$ such that with probability $1 - \delta$, $1/2\sum_i p_i \leq s \leq 2\sum_i p_i$ and the approximation $\tilde{\mathbf{K}} = \mathbf{KS}(\mathbf{SKS})^+\mathbf{SK}$ satisfies, for any rank $k$ orthogonal projection $\mathbf{X}$ and a positive constant $c$ independent of $\mathbf{X}$:*

$$\operatorname{tr}(\mathbf{K} - \mathbf{XKX}) \leq \operatorname{tr}(\tilde{\mathbf{K}} - \mathbf{X}\tilde{\mathbf{K}}\mathbf{X}) + c \leq (1 + \epsilon)\operatorname{tr}(\mathbf{K} - \mathbf{XKX}). \tag{16}$$

*When ridge leverage scores are computed exactly, $\sum_i p_i = O\left(\frac{k}{\epsilon}\log\frac{k}{\delta\epsilon}\right)$.*

Intuitively, Theorem 12 ensures that the distance from $\tilde{\mathbf{K}}$ to any low dimensional subspace closely approximates the distance from $\mathbf{K}$ to the subspace. Accordingly, $\tilde{\mathbf{K}}$ can be used in place of $\mathbf{K}$ to approximately solve low-rank approximation problems, both constrained (e.g. $k$-means clustering) and unconstrained (e.g. principal component analysis). See Theorems 17 and 18.

*Proof.* Set $c = \mathrm{tr}(\mathbf{K}) - \mathrm{tr}(\tilde{\mathbf{K}})$, which is $\geq 0$ since $\tilde{\mathbf{K}} \preceq \mathbf{K}$ by Theorem 3. By linearity of trace:

$$\mathrm{tr}(\tilde{\mathbf{K}} - \mathbf{X}\tilde{\mathbf{K}}\mathbf{X}) + c = \mathrm{tr}(\mathbf{K}) - \mathrm{tr}(\mathbf{X}\tilde{\mathbf{K}}\mathbf{X}).$$

So to obtain (16) it suffices to show:

$$\mathrm{tr}(\mathbf{X}\mathbf{K}\mathbf{X}) - \epsilon\,\mathrm{tr}(\mathbf{K} - \mathbf{X}\mathbf{K}\mathbf{X}) \leq \mathrm{tr}(\mathbf{X}\tilde{\mathbf{K}}\mathbf{X}) \leq \mathrm{tr}(\mathbf{X}\mathbf{K}\mathbf{X}). \tag{17}$$

Since $\mathbf{X}$ is a rank $k$ orthogonal projection we can write $\mathbf{X} = \mathbf{Q}\mathbf{Q}^T$ where $\mathbf{Q} \in \mathbb{R}^{n \times k}$ has orthonormal columns. Applying the cyclic property of the trace, and the spectral bound of Theorem 3:

$$\mathrm{tr}(\mathbf{X}\tilde{\mathbf{K}}\mathbf{X}) = \mathrm{tr}(\mathbf{Q}^T\tilde{\mathbf{K}}\mathbf{Q}) = \sum_{i=1}^{k} \mathbf{q}_i^T \tilde{\mathbf{K}}\mathbf{q}_i \leq \sum_{i=1}^{k} \mathbf{q}_i^T \mathbf{K}\mathbf{q}_i = \mathrm{tr}(\mathbf{Q}^T\mathbf{K}\mathbf{Q}) = \mathrm{tr}(\mathbf{X}\mathbf{K}\mathbf{X}).$$

This gives us the upper bound of (17). For the lower bound we apply Corollary 4:

$$\mathrm{tr}(\mathbf{X}\tilde{\mathbf{K}}\mathbf{X}) = \sum_{i=1}^{k} \mathbf{q}_i^T \tilde{\mathbf{K}}\mathbf{q}_i \geq \sum_{i=1}^{k} \mathbf{q}_i^T \mathbf{K}\mathbf{q}_i - k\epsilon\lambda = \mathrm{tr}(\mathbf{X}\mathbf{K}\mathbf{X}) - k\epsilon\lambda. \tag{18}$$

Finally, $k\epsilon\lambda = \epsilon \sum_{i=k+1}^{n} \sigma_i(\mathbf{K}) \leq \epsilon\,\mathrm{tr}(\mathbf{K} - \mathbf{X}\mathbf{K}\mathbf{X})$ since $\mathrm{tr}(\mathbf{K}) = \sum_{i=1}^{n} \sigma_i(\mathbf{K})$ and $\mathrm{tr}(\mathbf{X}\mathbf{K}\mathbf{X}) \leq \sum_{i=1}^{k} \sigma_i(\mathbf{K})$ by the Eckart-Young theorem. Plugging into (18) gives (17), completing the proof.

We conclude by showing that $s$ is not too large. As in the proof of Theorem 3, $s \leq 2\sum_i p_i$ with probability $1 - \delta$. When ridge leverage scores are computed exactly $\sum_i p_i \leq 16 \sum l_i^\lambda \log(\sum l_i^\lambda / \delta)$.

$$\begin{aligned}
\sum_i l_i^\lambda &= \mathrm{tr}(\mathbf{K}(\mathbf{K} + \epsilon \left(\frac{1}{k} \sum_{i=k+1}^{n} \sigma_i(\mathbf{K})\right) \mathbf{I})^{-1}) \\
&\leq \frac{1}{\epsilon} \mathrm{tr}(\mathbf{K}(\mathbf{K} + \left(\frac{1}{k} \sum_{i=k+1}^{n} \sigma_i(\mathbf{K})\right) \mathbf{I})^{-1}) \\
&= \frac{1}{\epsilon} \sum_{i=1}^{n} \frac{\sigma_i(\mathbf{K})}{\sigma_i(\mathbf{K}) + \frac{1}{k}\sum_{i=k+1}^{n} \sigma_i(\mathbf{K})} \\
&= \frac{1}{\epsilon} \left( \sum_{i=1}^{k} \frac{\sigma_i(\mathbf{K})}{\sigma_i(\mathbf{K}) + \frac{1}{k}\sum_{i=k+1}^{n} \sigma_i(\mathbf{K})} + \sum_{i=k+1}^{n} \frac{\sigma_i(\mathbf{K})}{\sigma_i(\mathbf{K}) + \frac{1}{k}\sum_{i=k+1}^{n} \sigma_i(\mathbf{K})} \right) \\
&\leq \frac{1}{\epsilon} \left( k + \sum_{i=k+1}^{n} \frac{\sigma_i(\mathbf{K})}{\frac{1}{k}\sum_{i=k+1}^{n} \sigma_i(\mathbf{K})} \right) = \frac{2k}{\epsilon}. \tag{19}
\end{aligned}$$

Accordingly, $\sum_i p_i = 32 \frac{k}{\epsilon} \log \frac{k}{\delta\epsilon}$ as desired. $\qquad\square$

## C   Correctness of Recursive RLS-Nyström Algorithm

In this section we prove Theorem 8, our main recursive invariant for proving the correctness of Algorithm 2, RECURSIVERLS-NYSTRÖM.

**Theorem 8.** *For any inputs $\mathbf{x}_1, \ldots, \mathbf{x}_m$, $K$, $\lambda > 0$ and $\delta \in (0, 1/32)$, let $\mathbf{K}$ be the kernel matrix for $\mathbf{x}_1, \ldots, \mathbf{x}_m$ and kernel function $K$ and let $d_{\mathrm{eff}}^\lambda(\mathbf{K})$ be the effective dimension of $\mathbf{K}$ with parameter $\lambda$. With probability $(1 - 3\delta)$, RECURSIVERLS-NYSTRÖM outputs $\mathbf{S}$ with $s$ columns that satisfies:*

$$\frac{1}{2}(\mathbf{B}^T\mathbf{B} + \lambda\mathbf{I}) \preceq (\mathbf{B}^T\mathbf{S}\mathbf{S}^T\mathbf{B} + \lambda\mathbf{I}) \preceq \frac{3}{2}(\mathbf{B}^T\mathbf{B} + \lambda\mathbf{I}) \qquad \textit{for any } \mathbf{B} \textit{ with } \mathbf{B}\mathbf{B}^T = \mathbf{K}. \tag{20}$$

*Additionally, $s \leq s_{\max}(d_{\mathrm{eff}}^\lambda(\mathbf{K}), \delta)$ where $s_{\max}(w, z) \stackrel{\text{def}}{=} 384 \cdot (w + 1) \log\left((w + 1)/z\right)$. The algorithm uses $\leq c_1 m s_{\max}(d_{\mathrm{eff}}^\lambda(\mathbf{K}), \delta)$ kernel evaluations and $\leq c_2 m s_{\max}(d_{\mathrm{eff}}^\lambda(\mathbf{K}), \delta)^2$ additional computation time where $c_1$ and $c_2$ are fixed universal constants.*

*Proof.* RECURSIVERLS-NYSTRÖM is a recursive algorithm and we prove Theorem 8 via induction on the size of the input, $m$. In particular, we will show that, if Theorem 8 holds for any all $m < n$, then it also holds for $m = n$. Our base case is $m = 1$.

**Base case: Theorem 8 holds for any inputs as long as $m = 1$.**

Suppose $m = 1$, so the input consists of a single point $\mathbf{x}_1$. Then the if statement on Line 1 clearly evaluates to true since $192 \log(1/\delta) > 1$. So, $\mathbf{S}$ is set to a $1 \times 1$ identity matrix and (20) of Theorem 8 holds trivially since $(\mathbf{B}^T\mathbf{B} + \lambda\mathbf{I}) = (\mathbf{B}^T\mathbf{S}\mathbf{S}^T\mathbf{B} + \lambda\mathbf{I})$. Furthermore, $s = 1 \leq s_{\max}(d_{\text{eff}}^\lambda(\mathbf{K}), \delta)$ for any $d_{\text{eff}}^\lambda(\mathbf{K})$ and $\delta$, as required. The algorithm runs in $O(1)$ time and performs no kernel evaluations, so the runtime requirements of Theorem 8 also hold as long as $c_2$ set to a large enough constant. This all holds with probability 1, and thus for any input failure probability $\delta$.

**Inductive Step: Theorem 8 holds for $m = n$ as long as it holds for all $m < n$.**

Depending on the setting of $\delta$, we split our analysis into 2 cases:

**Case 1: The number of input data points $n$ is $< 192 \log(1/\delta)$.**

In this case, as for the base case, the if statement on Line 1 evaluates to true. $\mathbf{S}$ is set to an $n \times n$ identity matrix so (6) holds trivially. Furthermore, the number of samples $s$ is equal to $n$, and $n < 192 \log(1/\delta) \leq s_{\max}(d_{\text{eff}}^\lambda(\mathbf{K}), \delta)$ as required. Again the algorithm doesn't compute any kernel dot products, the runtime bound required by Theorem 8 holds, and all statements hold with probability 1, which is $> 1 - 3\delta$ for any input failure probability $\delta$.

**Case 2: The number of input data points $n$ is $\geq 192 \log(1/\delta)$.**

For this case we will use our inductive assumption since RECURSIVERLS-NYSTRÖM will call itself recursively at Step 5, for a smaller input size $m < n$.

We first note that the expected number of samples taken in Step 4 is $n/2$. I.e. $\mathbb{E}|\bar{\mathcal{S}}| = n/2$. By a standard multiplicative error Chernoff bound, with high probability the number of samples taken is not much larger than this expectation. This is important because it tells us that our problem size decreases substantially before we make the recursive call in Step 5. Following the simplified Chernoff bounds in e.g. [MU17], when $n \geq 192 \log(1/\delta)$, and thus $\mathbb{E}|\bar{\mathcal{S}}| \geq 96 \log(1/\delta)$, we have :

$$\mathbb{P}\left[1 \leq |\bar{\mathcal{S}}| \leq .56n\right] \geq (1 - \delta) \tag{21}$$

as long as $\delta < 1/32$, as required by Theorem 8.

So, with probability $(1 - \delta)$, on Step (5), RECURSIVERLS-NYSTRÖM is called recursively on a data set $\bar{\mathbf{X}}$ of size $\geq 1$ and $\leq .56n$. Accordingly, we can apply our inductive assumption that Theorem 8 holds for all $m$ between 1 and $n - 1$ to conclude that, with probability $(1 - 3 \cdot \delta/3)$[2]:

1. Let $\mathbf{K}_{\bar{\mathcal{S}}}$ denote the kernel matrix for the data points in $\bar{\mathbf{X}}$ (corresponding to the sample $\bar{\mathcal{S}}$ with kernel function $K$. Then $\mathbf{B}_{\bar{\mathcal{S}}} = \bar{\mathbf{S}}^T\mathbf{B}$ satisfies $\mathbf{B}_{\bar{\mathcal{S}}}\mathbf{B}_{\bar{\mathcal{S}}}^T = \mathbf{K}_{\bar{\mathcal{S}}}$. Thus:

$$\frac{1}{2}(\mathbf{B}_{\bar{\mathcal{S}}}^T\mathbf{B}_{\bar{\mathcal{S}}} + \lambda\mathbf{I}) \preceq (\mathbf{B}_{\bar{\mathcal{S}}}^T\tilde{\mathbf{S}}\tilde{\mathbf{S}}^T\mathbf{B}_{\bar{\mathcal{S}}} + \lambda\mathbf{I}) \preceq \frac{3}{2}(\mathbf{B}_{\bar{\mathcal{S}}}^T\mathbf{B}_{\bar{\mathcal{S}}} + \lambda\mathbf{I}). \tag{22}$$

2. $\tilde{\mathbf{S}}$ has $\leq s_{\max}(d_{\text{eff}}^\lambda(\mathbf{K}_{\bar{\mathcal{S}}}), \delta/3)$ columns.

3. The recursive call at Step 5 evaluates $K$, the kernel function, $\leq c_1 \cdot |\bar{\mathcal{S}}| \cdot s_{\max}(d_{\text{eff}}^\lambda(\mathbf{K}_{\bar{\mathcal{S}}}), \delta/3)$ times and uses $\leq c_2 \cdot |\bar{\mathcal{S}}| \cdot s_{\max}(d_{\text{eff}}^\lambda(\mathbf{K}_{\bar{\mathcal{S}}}), \delta/3)^2$ additional runtime steps.

We first use (22) to prove (20). We can write $\mathbf{K}_{\bar{\mathcal{S}}} = \bar{\mathbf{S}}^T\mathbf{K}\bar{\mathbf{S}}$. For all $i \in \{1, \ldots n\}$ let

$$\bar{\ell}_i^\lambda = \left(\mathbf{B}\left(\mathbf{B}^T\bar{\mathbf{S}}\bar{\mathbf{S}}^T\mathbf{B} + \lambda\mathbf{I}\right)^{-1}\mathbf{B}^T\right)_{i,i} \qquad \text{and} \qquad \bar{p}_i = \min\{1, 16\bar{l}_i^\lambda \log(\sum_i \bar{l}_i^\lambda/\delta)\}.$$

By Lemma 5, since $\bar{\mathbf{S}}$ is constructed by sampling with probability $\frac{1}{2}$, with probability $1 - \delta$, for all $i$:

$$\bar{\ell}_i^\lambda \geq \ell_i^\lambda(\mathbf{K}) \qquad \text{and} \qquad \sum_{i=1}^n \bar{p}_i \leq 64 d_{\text{eff}}^\lambda(\mathbf{K}) \log\left(d_{\text{eff}}^\lambda(\mathbf{K})/\delta\right). \qquad (23)$$

Here $\ell_i^\lambda(\mathbf{K})$ is the exact $i^{\text{th}}$ $\lambda$-ridge leverage score of $\mathbf{K}$.

Now, since $\mathbf{B}_{\bar{\mathcal{S}}} = \bar{\mathbf{S}}^T \mathbf{B}$, it follows from (22) and from the well known fact that $\mathbf{M} \preceq \mathbf{N} \implies \mathbf{N}^{-1} \preceq \mathbf{M}^{-1}$, that for any vector $\mathbf{z}$,

$$\frac{2}{3}\mathbf{z}^T \left(\mathbf{B}^T \bar{\mathbf{S}}\bar{\mathbf{S}}^T \mathbf{B} + \lambda \mathbf{I}\right)^{-1} \mathbf{z} \leq \mathbf{z}^T \left(\mathbf{B}^T \bar{\mathbf{S}}\tilde{\mathbf{S}}\tilde{\mathbf{S}}^T\bar{\mathbf{S}}^T \mathbf{B} + \lambda \mathbf{I}\right)^{-1} \mathbf{z} \leq 2\mathbf{z}^T \left(\mathbf{B}^T \bar{\mathbf{S}}\bar{\mathbf{S}}^T \mathbf{B} + \lambda \mathbf{I}\right)^{-1} \mathbf{z}.$$

Accordingly, since we set $\hat{\mathbf{S}} := \bar{\mathbf{S}} \cdot \tilde{\mathbf{S}}$, for all $i \in \{1, \ldots, n\}$

$$\bar{\ell}_i^\lambda \leq \frac{3}{2} \left(\mathbf{B}\left(\mathbf{B}^T \hat{\mathbf{S}}\hat{\mathbf{S}}^T \mathbf{W}\mathbf{B} + \lambda \mathbf{I}\right)^{-1} \mathbf{B}^T\right)_{i,i} \leq 3\bar{\ell}_i^\lambda. \qquad (24)$$

By Lemma 6, the middle term is exactly equal to $\tilde{l}_i^\lambda$ as computed in Step 7 of RECURSIVERLS-NYSTRÖM. So combining (24) and (23) we have that:

$$\tilde{\ell}_i^\lambda \geq \ell_i^\lambda(\mathbf{K}) \qquad \text{and} \qquad \sum_{i=1}^n p_i \leq 192 d_{\text{eff}}^\lambda(\mathbf{K}) \log\left(d_{\text{eff}}^\lambda(\mathbf{K})/\delta\right). \qquad (25)$$

The second bound holds because, as computed on Step 8 of RECURSIVERLS-NYSTRÖM,

$$p_i = \min\{1, \tilde{l}_i^\lambda \cdot 16 \log(\textstyle\sum \tilde{l}_i^\lambda/\delta)\} \leq 3\min\{1, \bar{l}_i^\lambda \cdot 16 \log(\textstyle\sum \bar{l}_i^\lambda/\delta)\}$$
$$= 3\bar{p}_i \leq 192 d_{\text{eff}}^\lambda(\mathbf{K}) \log\left(d_{\text{eff}}^\lambda(\mathbf{K})/\delta\right)$$

by (24). Equation (25) guarantees that $\mathbf{S}$ is sampled by valid over-estimates of the ridge leverage scores and we have a bound on the sum of the sampling probabilities. So, to establish (20), we just apply the matrix Bernstein results presented in Lemma 9. We conclude that, with probability $(1 - \delta)$,

$$\frac{1}{2}(\mathbf{B}^T \mathbf{B} + \lambda \mathbf{I}) \preceq (\mathbf{B}^T \mathbf{S}\mathbf{S}^T \mathbf{B} + \lambda \mathbf{I}) \preceq \frac{3}{2}(\mathbf{B}^T \mathbf{B} + \lambda \mathbf{I}) \qquad \text{for any } \mathbf{B} \text{ with } \mathbf{B}\mathbf{B}^T = \mathbf{K}.$$

The same lemma guarantees that $\mathbf{S}$ will have $s$ columns where

$$\frac{1}{2} \sum p_i \leq s \leq 2 \sum p_i. \qquad (26)$$

$2 \sum p_i \leq 384 d_{\text{eff}}^\lambda(\mathbf{K}) \log\left(d_{\text{eff}}^\lambda(\mathbf{K})/\delta\right) \leq s_{\max}(d_{\text{eff}}^\lambda(\mathbf{K}), \delta)$ columns.

To finish our proof of Theorem 8, we still need a bound the algorithms runtime.

Kernel evaluations are performed both during the recursive call at Step 5 and when computing approximate leverage scores at Step 7. Let $\tilde{s}$ be the number of columns in $\tilde{\mathbf{S}}$, and hence in $\hat{\mathbf{S}}$. At Step 7, $K$ needs to be evaluated $n \cdot (\tilde{s} + 1)$ times: $n\tilde{s}$ times to compute $\mathbf{K}\hat{\mathbf{S}}$ and $n$ times to compute the diagonal of $\mathbf{K}$. Additionally, by the 3rd guarantee that comes from our inductive assumption, we need at most $c_1 \cdot |\bar{\mathcal{S}}| \cdot s_{\max}(d_{\text{eff}}^\lambda(\mathbf{K}_{\bar{\mathcal{S}}}), \delta/3)$ kernel evaluations for the recursive call. We claim that:

$$s_{\max}(d_{\text{eff}}^\lambda(\mathbf{K}_{\bar{\mathcal{S}}}), \delta/3) \leq 1.317 s_{\max}(d_{\text{eff}}^\lambda(\mathbf{K}), \delta). \qquad (27)$$

This follows from Lemma 20: since $\mathbf{K}_{\bar{\mathcal{S}}} = \bar{\mathbf{S}}^T \mathbf{K}\bar{\mathbf{S}}$ and $\bar{\mathbf{S}}\bar{\mathbf{S}}^T \preceq \mathbf{I}$ for any sampling matrix, $d_{\text{eff}}^\lambda(\mathbf{K}_{\bar{\mathcal{S}}}) \leq d_{\text{eff}}^\lambda(\mathbf{K})$. Additionally, we use that $\log(3/\delta) \leq 1.317 \log(1/\delta)$ when $\delta \leq 1/32$.

Using this bound and (21) we see that our total number of kernel evaluations can be bounded by:

$$n \cdot (\tilde{s} + 1) + c_1 \cdot |\bar{\mathcal{S}}| \cdot s_{\max}(d_{\text{eff}}^\lambda(\mathbf{K}_{\bar{\mathcal{S}}}), \delta/3)$$
$$\leq n \cdot (s_{\max}(d_{\text{eff}}^\lambda(\mathbf{K}_{\bar{\mathcal{S}}}), \delta/3) + 1) + c_1 \cdot .56n \cdot s_{\max}(d_{\text{eff}}^\lambda(\mathbf{K}_{\bar{\mathcal{S}}}), \delta/3)$$
$$\leq (2.317 + .74c_1)\, n \cdot s_{\max}(d_{\text{eff}}^\lambda(\mathbf{K}), \delta).$$

As long as $c_1 > 9$, the above is $< c_1 n s_{\max}(d_{\text{eff}}^\lambda(\mathbf{K}), \delta)$, so we see that RECURSIVERLS-NYSTRÖM run on a data set of size $n$ performs no more kernel evaluations than that allowed by Theorem 8.

We finally bound runtime, accounting for the recursive call to RECURSIVERLS-NYSTRÖM and all other steps. Again, using the 3rd guarantee from our inductive assumption, (27), and (21) to bound $|\bar{\mathcal{S}}|$, the recursive call that computes $\tilde{\mathbf{S}}$ has runtime at most:

$$c_2 \cdot |\bar{\mathcal{S}}| \cdot s_{\max}(d_{\text{eff}}^\lambda(\mathbf{K}_{\bar{\mathcal{S}}}), \delta/3)^2 \leq .972 c_2 n \cdot s_{\max}(d_{\text{eff}}^\lambda(\mathbf{K}), \delta)^2.$$

In addition to the recursive call, the remaining runtime of the algorithm is dominated by the time to compute $\left(\hat{\mathbf{S}}^T \mathbf{K} \hat{\mathbf{S}} + \lambda \mathbf{I}\right)^{-1}$ and then to multiply this matrix by the $n \times \tilde{s}$ matrix $\mathbf{K}\hat{\mathbf{S}}$ at Step 7. Both of these operations and all other steps can be performed in $O(\tilde{s}^3 + n\tilde{s}^2)$ time. Since $\tilde{s} \leq n$, there is some constant $c$ such that the number of steps required for the algorithm besides the recursive call is $cn\tilde{s}^2 \leq cn s_{\max}(d_{\text{eff}}^\lambda(\mathbf{K}_{\bar{\mathcal{S}}}), \delta/3)^2$. Again applying (27), our total runtime is bounded by:

$$.972 c_2 n \cdot s_{\max}(d_{\text{eff}}^\lambda(\mathbf{K}), \delta)^2 + cn s_{\max}(d_{\text{eff}}^\lambda(\mathbf{K}_{\bar{\mathcal{S}}}), \delta/3)^2$$

which is $\leq c_2 n \cdot s_{\max}(d_{\text{eff}}^\lambda(\mathbf{K}), \delta)^2$ as long as $c_2 \geq 40c$.

The proof of our statements above relied on three events succeeding: (21), (23), that the recursive call succeeded in satisfying (22) and the two following guarantees. Each of these events fails with probability at most $\delta$, so we conclude via a union bound that they all succeed with probability $1 - 3\delta$.

Accordingly, we have proven that Theorem (8) holds for fixed universal constants $c_1$ and $c_2$ for any input data set of size $n$ as long as it holds for any input data set of size $m$ with $1 \leq m < n$. Along with our base case, this establishes the theorem for all input sizes. $\qquad\square$

## D   Recursive RLS-Nyström algorithm for fixed sample size

We now discuss our variant of Algorithm 2 where, instead of fixing $\lambda$, the user sets a sample size $s$ and $\lambda$ is determined such that $s = \Theta(d_{\text{eff}}^\lambda \cdot \log(d_{\text{eff}}^\lambda/\delta))$. This variant is practically useful and necessary in applications to kernel PCA and kernel $k$-means clustering, when $\lambda$ is unknown, but where we set $s \approx k \log k$ (see Appendices B and E).

Given a fixed sample size $s$, we will control $\lambda$ using the following fact:

**Fact 13** (Proven in (19))**.** *For any $\mathbf{K}$ and integer $k$, for $\lambda = \frac{1}{k} \sum_{i=k+1}^n \sigma_i(\mathbf{K})$, $d_{\text{eff}}^\lambda \leq 2k$.*

If we choose $k$ such that $s \approx k \log k$ then setting $\lambda$ as above will yield an RLS-Nyström approximation with approximately $s$ sampled columns. The details are given in Algorithm 3.

---

**Algorithm 3** RECURSIVE RLS-NYSTRÖM SAMPLING, FIXED SAMPLE SIZE.

---

**input**: $\mathbf{x}_1, \ldots, \mathbf{x}_m \in \mathcal{X}$, kernel function $K : \mathcal{X} \times \mathcal{X} \to \mathbb{R}$, sample size $s$, failure prob. $\delta \in (0, 1/32)$
**output**: sampling matrix $\mathbf{S} \in \mathbb{R}^{m \times s'}$.
1: **if** $m \leq s$ **then**
2:     **return** $\mathbf{S} := \mathbf{I}_{m \times m}$.
3: **end if**
4: Let $\bar{\mathcal{S}}$ be a random subset of $\{1, ..., m\}$, with each $i$ included independently with probability $\frac{1}{2}$.
    $\triangleright$ Let $\bar{\mathbf{X}} = \{\mathbf{x}_{i_1}, \mathbf{x}_{i_2}, ..., \mathbf{x}_{i_{|\bar{\mathcal{S}}|}}\}$ for $i_j \in \bar{\mathcal{S}}$ be the data sample corresponding to $\bar{\mathcal{S}}$.
    $\triangleright$ Let $\bar{\mathbf{S}} = [\mathbf{e}_{i_1}, \mathbf{e}_{i_2}, ..., \mathbf{e}_{i_{|\bar{\mathcal{S}}|}}]$ be the sampling matrix corresponding to $\bar{\mathcal{S}}$.
5: $\tilde{\mathbf{S}} := \text{RECURSIVERLS-NYSTRÖM}(\bar{\mathbf{X}}, K, s, \delta/3)$.
6: $\hat{\mathbf{S}} := \bar{\mathbf{S}} \cdot \tilde{\mathbf{S}}$.
7: Set $k$ to the maximum integer with $ck \log(2k/\delta) \leq s$, where $c$ is some fixed constant.
8: $\tilde{\lambda} := \frac{1}{k} \sum_{i=k+1}^n \sigma_i(\hat{\mathbf{S}}^T \mathbf{K} \hat{\mathbf{S}})$                                             $\triangleright$ Approximate $\lambda$
9: Set $\tilde{l}_i^\lambda := \frac{5}{\tilde{\lambda}} \left( \mathbf{K} - \mathbf{K}\hat{\mathbf{S}} \left( \hat{\mathbf{S}}^T \mathbf{K} \hat{\mathbf{S}} + \tilde{\lambda} \mathbf{I} \right)^{-1} \hat{\mathbf{S}}^T \mathbf{K} \right)_{i,i}$ for each $i \in \{1, ..., m\}$.

    $\triangleright$ By Lemma 6, equals $\frac{3}{2} (\mathbf{B}(\mathbf{B}^T \hat{\mathbf{S}} \hat{\mathbf{S}}^T \mathbf{B} + \tilde{\lambda} \mathbf{I})^{-1} \mathbf{B}^T)_{i,i}$. $\mathbf{K}$ denotes the kernel matrix for datapoints $\{\mathbf{x}_1, \ldots, \mathbf{x}_m\}$ and kernel function $K$.
10: Set $p_i := \min\{1, \tilde{l}_i^\lambda \cdot 16 \log(2k/\delta)\}$ for each $i \in \{1, ..., m\}$.
11: Initially set weighted sampling matrix $\mathbf{S}$ to be empty. For each $i \in \{1, \ldots, m\}$, with probability $p_i$, append the column $\frac{1}{\sqrt{p_i}} \mathbf{e}_i$ onto $\mathbf{S}$.
12: **return** $\mathbf{S}$

---

**Theorem 14.** *For sufficiently large universal constant $c$, let $k$ be any positive integer with $s \geq ck \log(2k/\delta)$ and $\lambda = \frac{1}{k} \sum_{i=k+1}^{n} \sigma_i(\mathbf{K})$. Let $\mathbf{S} \in \mathbb{R}^{n \times s'}$ be computed by Algorithm 3. With probability $1 - 3\delta$, $s' \leq 2s$, $\mathbf{S}$ is sampled by overestimates of the $\lambda$-ridge leverage scores of $\mathbf{K}$, and the Nyström approximation $\tilde{\mathbf{K}} = \mathbf{K}\mathbf{S}(\mathbf{S}^T\mathbf{K}\mathbf{S})^+\mathbf{S}^T\mathbf{K}$ satisfies the guarantee of Theorem 3. Algorithm 3 uses $O(ns)$ kernel evaluations and $O(ns^2)$ runtime.*

For the $\lambda$ given in Theorem 14, we have $d_{\text{eff}}^{\lambda} = \Theta(k)$. Hence, since we set $s = \Theta(k \log k/\delta)$, additive error $\lambda$ is essentially the smallest we can obtain using an $s$ sample Nyström approximation. The proof of Theorem 14 is similar to that of Theorem 7. It follows from the recursive invariant:

**Theorem 15.** *With probability $1 - 3\delta$, Algorithm 3 performs $O(ns)$ kernel evaluations, runs in $O(ns^2)$ time, and for any integer $k$ with $s \geq ck \log(2k/\delta)$ returns $\mathbf{S}$ satisfying, for any $\mathbf{B}$ with $\mathbf{B}\mathbf{B}^T = \mathbf{K}$:*

$$\frac{1}{2}(\mathbf{B}^T\mathbf{B} + \lambda\mathbf{I}) \preceq (\mathbf{B}^T\mathbf{S}\mathbf{S}^T\mathbf{B} + \lambda\mathbf{I}) \preceq \frac{3}{2}(\mathbf{B}^T\mathbf{B} + \lambda\mathbf{I}) \tag{28}$$

*for $\lambda = \frac{1}{k} \sum_{i=k+1}^{n} \sigma_i(\mathbf{K})$.*

*Proof.* Assume by induction that after forming $\bar{\mathbf{S}}$ via uniformly sampling, the recursive call to Algorithm 3 returns $\tilde{\mathbf{S}}$ such that $\hat{\mathbf{S}} = \bar{\mathbf{S}} \cdot \tilde{\mathbf{S}}$ satisfies:

$$\frac{1}{2}(\mathbf{B}^T\bar{\mathbf{S}}\bar{\mathbf{S}}^T\mathbf{B} + \lambda'\mathbf{I}) \preceq (\mathbf{B}^T\hat{\mathbf{S}}\hat{\mathbf{S}}^T\mathbf{B} + \lambda'\mathbf{I}) \preceq \frac{3}{2}(\mathbf{B}^T\bar{\mathbf{S}}\bar{\mathbf{S}}^T\mathbf{B} + \lambda'\mathbf{I}). \tag{29}$$

where $\lambda' = \frac{1}{k} \sum_{i=k+1}^{n} \sigma_i(\bar{\mathbf{S}}^T\mathbf{K}\bar{\mathbf{S}})$. This implies that $\tilde{\lambda} = \frac{1}{k} \sum_{i=k+1}^{n} \sigma_i(\hat{\mathbf{S}}^T\mathbf{K}\hat{\mathbf{S}})$ satisfies:

$$\frac{1}{2k}\left(\sum_{i=k+1}^{n} \sigma_i(\bar{\mathbf{S}}^T\mathbf{K}\bar{\mathbf{S}}) + k\lambda'\right) \leq \tilde{\lambda} \leq \frac{3}{2k}\left(\sum_{i=k+1}^{n} \sigma_i(\bar{\mathbf{S}}^T\mathbf{K}\bar{\mathbf{S}}) + k\lambda'\right)$$

$$\lambda' \leq \tilde{\lambda} \leq 3\lambda'.$$

Combining with (29) we have:

$$\frac{1}{2}(\mathbf{B}^T\bar{\mathbf{S}}\bar{\mathbf{S}}^T\mathbf{B} + \lambda'\mathbf{I}) \preceq (\mathbf{B}^T\hat{\mathbf{S}}\hat{\mathbf{S}}^T\mathbf{B} + \tilde{\lambda}\mathbf{I}) \preceq \frac{9}{2}(\mathbf{B}^T\bar{\mathbf{S}}\bar{\mathbf{S}}^T\mathbf{B} + \lambda'\mathbf{I}).$$

So, for all $i$, $\tilde{l}_i^{\lambda}$ (which is computed using $(\mathbf{B}^T\hat{\mathbf{S}}\hat{\mathbf{S}}^T\mathbf{B} + \tilde{\lambda}\mathbf{I})$ and oversampling factor 5 in Step 9 of Algorithm 3) is at least as large as the approximate leverage score computed using $\bar{\mathbf{S}}$ instead of $\hat{\mathbf{S}}$. If we sample by these scores, by Lemma 5 and Lemma 9 we will have with probability $1 - \delta$:

$$\frac{1}{2}(\mathbf{B}^T\mathbf{B} + \lambda'\mathbf{I}) \preceq (\mathbf{B}^T\mathbf{S}\mathbf{S}^T\mathbf{B} + \lambda'\mathbf{I}) \preceq \frac{3}{2}(\mathbf{B}^T\mathbf{B} + \lambda'\mathbf{I})$$

which implies (28) since $\lambda' \leq \lambda$ since $\|\bar{\mathbf{S}}\|_2 \leq 1$ so $\sigma_i(\bar{\mathbf{S}}^T\mathbf{K}\bar{\mathbf{S}}) \leq \sigma_i(\mathbf{K})$ for all $i$.

It just remains to show that we do not sample too many points. This can be shown using a similar reweighting argument to that used in the fixed $\lambda$ case in Lemma 5. Full details appear in Lemma 13 of [CMM17]. When forming the reweighting matrix $\mathbf{W}$, decreasing $\mathbf{W}_{i,i}$ will decrease $\sum_{i=k+1}^{n} \sigma_i(\mathbf{W}\mathbf{K}\mathbf{W})$ and hence will decrease $\lambda$. However, it is not hard to show that the $i^{\text{th}}$ ridge leverage score will still decrease. So we can find $\mathbf{W}$ giving a uniform ridge leverage score upper bound of $\alpha$. Let $\lambda' = \sum_{i=k+1}^{n} \sigma_i(\mathbf{W}\mathbf{K}\mathbf{W})$.

Using the same argument as Lemma 5, we can bound the sum of estimated sampling probabilities by $64 \log(\sum l_i^{\lambda'}(\mathbf{W}\mathbf{K}\mathbf{W})/\delta) \cdot \sum l_i^{\lambda'}(\mathbf{W}\mathbf{K}\mathbf{W}) \leq s/5$ by Fact 13 if we set $c$ large enough. The runtime and failure probability analysis is identical to that of Algorithm 2 (Theorem 8) – the only extra step is computing $\tilde{\lambda}$ which can be done in $O(s^3)$ time via an SVD of $\hat{\mathbf{S}}^T\mathbf{K}\hat{\mathbf{S}}$. $\square$

*Proof of Theorem 14.* The theorem follows immediately since Theorem 15 guarantees that in the final level of recussion $\mathbf{K}$ is sampled by overestimates of its $\lambda$-ridge leverage scores. The runtime bound follows from Theorem 15 and the fact that it is possible to compute $\mathbf{K}\mathbf{S}$ using $O(ns)$ kernel evaluations and $(\mathbf{S}^T\mathbf{K}\mathbf{S})^+$ using $O(ns^2 + s^3) = O(ns^2)$ additional time. $\square$

# E   Applications to learning tasks

In this section use our general approximation gaurantees from Theorems 3 and 12 to prove that the kernel approximations given by RLS-Nyström sampling are sufficient for many downstream learning tasks. In other words, $\tilde{\mathbf{K}}$ can be used in place of $\mathbf{K}$ without sacrificing accuracy or statistical performance in the final computation.

## E.1   Kernel ridge regression

We begin with a standard formulation of the ubiquitous kernel ridge regression task [SS02]. Given input data points $\mathbf{x}_1, \ldots, \mathbf{x}_n \in \mathbb{R}^d$ and labels $y_1, \ldots, y_n \in \mathbb{R}$ this problem asks us to solve:

$$\boldsymbol{\alpha} \overset{\text{def}}{=} \arg \min_{\mathbf{c} \in \mathbb{R}^n} \|\mathbf{Kc} - \mathbf{y}\|_2^2 + \lambda \mathbf{c}^T \mathbf{Kc}, \tag{30}$$

which can be done in closed form by computing:

$$\boldsymbol{\alpha} = (\mathbf{K} + \lambda \mathbf{I})^{-1} \mathbf{y}.$$

For prediction, when we're given a new input $\mathbf{x}$, we evaluate its label to be:

$$y = \sum_{i=1}^{n} \alpha_i K(\mathbf{x}_i, \mathbf{x}). \tag{31}$$

### E.1.1   Approximate kernel ridge regression

Naively, solving for $\boldsymbol{\alpha}$ exactly requires at least $O(n^2)$ time to compute $\mathbf{K}$, plus the cost of a direct or iterative matrix inversion algorithm. Prediction is also costly since it requires a kernel evaluation with all $n$ training points. These costs can be reduced significantly using Nyström approximation.

In particular, we first select landmark points and compute the kernel approximation $\tilde{\mathbf{K}} = \mathbf{KS}(\mathbf{S}^T \mathbf{KS})^+ \mathbf{S}^T \mathbf{K}$. We can then compute an approximate set of coefficients:

$$\tilde{\boldsymbol{\alpha}} \overset{\text{def}}{=} (\tilde{\mathbf{K}} + \lambda \mathbf{I})^{-1} \mathbf{y}. \tag{32}$$

With a direct matrix inversion, doing so only takes $O(ns^2)$ time when our sampling matrix $\mathbf{S} \in \mathbb{R}^{n \times s}$ selects $s$ landmark points. This is a significant improvement on the $O(n^3)$ time required to invert the full kernel. Additionally, the cost of multiplying by $\tilde{\mathbf{K}} + \lambda \mathbf{I}$, which determines the cost of most iterative regression solvers, is reduced, from $O(n^2)$ to $O(ns)$.

To predict a label for a new $\mathbf{x}$, we first compute its kernel product with all of our landmark points. Specifically, let $\mathbf{x}^{(1)}, \ldots, \mathbf{x}^{(s)}$ be the landmarks selected by $\mathbf{S}$'s columns. Define $\mathbf{w} \in \mathbb{R}^s$ as:

$$\mathbf{w}_i \overset{\text{def}}{=} K(\mathbf{x}^{(i)}, \mathbf{x}).$$

and let

$$y = \mathbf{w}^T (\mathbf{S}^T \mathbf{KS})^+ \mathbf{S}^T \mathbf{K} \tilde{\boldsymbol{\alpha}}. \tag{33}$$

Computationally, it makes sense to precompute $(\mathbf{S}^T \mathbf{KS})^+ \mathbf{S}^T \mathbf{K} \tilde{\boldsymbol{\alpha}}$. Then the cost of prediction is just $s$ kernel evaluations to compute $\mathbf{w}$, plus $s$ additional operations to multiply $\mathbf{w}^T$ by $(\mathbf{S}^T \mathbf{KS})^+ \mathbf{S}^T \mathbf{K} \tilde{\boldsymbol{\alpha}}$.

This approach is the standard way of applying Nyström approximation to the ridge regression problem and there are a number of ways to evaluate its performance. Beyond directly bounding minimization error for (30) (see e.g. [CLL$^+$15, YPW15, YZ13]), one particularly natural approach is to consider how the statistical risk of the estimator output by our approximate ridge regression routine compares to that of the exactly computed estimator.

### E.1.2   Relative error bound on statistical risk

To evaluate statistical risk we consider a *fixed design* setting which has been especially -popular [Bac13, AM15, LJS16, PD16]. Note that more complex statistical models can be analyzed as well

[HKZ14, RCR15]. In this setting, we assume that our observed labels $\mathbf{y} = [y_1, \ldots, y_n]$ represent underlying true labels $\mathbf{z} = [z_1, \ldots, z_n]$ perturbed with noise. For simplicity, we assume uniform Gaussian noise with variance $\sigma^2$, but more general noise models can be handled with essentially the same proof [Bac13]. In particular, our modeling assumption is that:

$$y_i = z_i + \eta_i$$

where $\eta_i \sim N(0, \sigma^2)$.

Following, [Bac13] and [AM15], we want to bound the expected in sample risk of our estimator for $\mathbf{z}$, which is computed using the noisy measurements $\mathbf{y} = \mathbf{z} + \boldsymbol{\eta}$. For exact kernel ridge regression, we can check from (31) that this estimator is equal to $\mathbf{K}\boldsymbol{\alpha}$. The risk $\mathcal{R}$ is:

$$\mathcal{R} \overset{\text{def}}{=} \mathbb{E}_{\boldsymbol{\eta}} \|\mathbf{K}(\mathbf{K} + \lambda\mathbf{I})^{-1}(\mathbf{z} + \boldsymbol{\eta}) - \mathbf{z}\|_2^2$$
$$= \|\left(\mathbf{K}(\mathbf{K} + \lambda\mathbf{I})^{-1} - \mathbf{I}\right)\mathbf{z}\|_2^2 + \mathbb{E}_{\boldsymbol{\eta}} \|\mathbf{K}(\mathbf{K} + \lambda\mathbf{I})^{-1}\boldsymbol{\eta}\|_2^2$$
$$= \lambda^2 \mathbf{z}^T(\mathbf{K} + \lambda\mathbf{I})^{-2}\mathbf{z} + \sigma^2 \operatorname{tr}(\mathbf{K}^2(\mathbf{K} + \lambda\mathbf{I})^{-2}).$$

The two terms that compose $\mathcal{R}$ are referred to as the bias and variance terms of the risk:

$$\operatorname{bias}(\mathbf{K})^2 \overset{\text{def}}{=} \lambda^2 \mathbf{z}^T(\mathbf{K} + \lambda\mathbf{I})^{-2}\mathbf{z}$$

$$\operatorname{variance}(\mathbf{K}) \overset{\text{def}}{=} \sigma^2 \operatorname{tr}(\mathbf{K}^2(\mathbf{K} + \lambda\mathbf{I})^{-2}).$$

For approximate kernel ridge regression, it follows from (33) that our predictor for $\mathbf{z}$ is $\tilde{\mathbf{K}}\tilde{\boldsymbol{\alpha}}$. Accordingly, the risk of the approximate estimator, $\tilde{\mathcal{R}}$ is equal to:

$$\tilde{\mathcal{R}} = \operatorname{bias}(\tilde{\mathbf{K}})^2 + \operatorname{variance}(\tilde{\mathbf{K}})$$

We're are ready to prove our main theorem on kernel ridge regression.

**Theorem 16** (Kernel Ridge Regression Risk Bound). *Suppose $\tilde{\mathbf{K}}$ is computed using RLS-Nyström with approximation parameter $\epsilon\lambda$ and failure probability $\delta \in (0, 1/8)$. Let $\tilde{\boldsymbol{\alpha}} = (\tilde{\mathbf{K}} + \lambda\mathbf{I})^{-1}\mathbf{y}$ and let $\tilde{\mathbf{K}}\tilde{\boldsymbol{\alpha}}$ be our estimator for $\mathbf{z}$ computed with the approximate kernel. With probability $1 - \delta$:*

$$\tilde{\mathcal{R}} \le (1 + 3\epsilon)\mathcal{R}.$$

*By Theorem 7, Algorithm 2 can compute $\tilde{\mathbf{K}}$ with just $O(ns)$ kernel evaluations and $O(ns^2)$ computation time, with $s = O\left(\frac{d_{\text{eff}}^\lambda}{\epsilon} \log \frac{d_{\text{eff}}^\lambda}{\delta\epsilon}\right)$.*

In other words, replacing $\mathbf{K}$ with the approximation $\tilde{\mathbf{K}}$ is provably sufficient for obtaining a $1 + \Theta(\epsilon)$ quality solution to the downstream task of ridge regression.

*Proof.* The proof follows that of Theorem 1 in [AM15]. First we show that:

$$\operatorname{bias}(\tilde{\mathbf{K}}) \le (1 + \epsilon)\operatorname{bias}(\mathbf{K}). \tag{34}$$

At first glance this might appear trivial as Theorem 3 easily implies that

$$(\tilde{\mathbf{K}} + \lambda\mathbf{I})^{-1} \preceq (1 + \epsilon)(\mathbf{K} + \lambda\mathbf{I})^{-1}$$

However, this statement *does not imply* that

$$(\tilde{\mathbf{K}} + \lambda\mathbf{I})^{-2} \preceq (1 + \epsilon)^2(\mathbf{K} + \lambda\mathbf{I})^{-2}$$

since $(\tilde{\mathbf{K}} + \lambda\mathbf{I})^{-1}$ and $(\mathbf{K} + \lambda\mathbf{I})^{-1}$ do not necessarily commute. Instead we proceed:

$$\frac{1}{\lambda}\operatorname{bias}(\tilde{\mathbf{K}}) = \|(\tilde{\mathbf{K}} + \lambda\mathbf{I})^{-1}\mathbf{z}\|_2$$
$$\le \|(\mathbf{K} + \lambda\mathbf{I})^{-1}\mathbf{z}\|_2 + \|(\tilde{\mathbf{K}} + \lambda\mathbf{I})^{-1}\mathbf{z} - (\mathbf{K} + \lambda\mathbf{I})^{-1}\mathbf{z}\|_2 \qquad \text{(triangle inequality)}$$
$$= \|(\mathbf{K} + \lambda\mathbf{I})^{-1}\mathbf{z}\|_2 + \|(\tilde{\mathbf{K}} + \lambda\mathbf{I})^{-1}[(\mathbf{K} + \lambda\mathbf{I}) - (\tilde{\mathbf{K}} + \lambda\mathbf{I})](\mathbf{K} + \lambda\mathbf{I})^{-1}\mathbf{z}\|_2$$
$$= \|(\mathbf{K} + \lambda\mathbf{I})^{-1}\mathbf{z}\|_2 + \|(\tilde{\mathbf{K}} + \lambda\mathbf{I})^{-1}(\mathbf{K} - \tilde{\mathbf{K}})(\mathbf{K} + \lambda\mathbf{I})^{-1}\mathbf{z}\|_2$$
$$\le \|(\mathbf{K} + \lambda\mathbf{I})^{-1}\mathbf{z}\|_2 + \|(\tilde{\mathbf{K}} + \lambda\mathbf{I})^{-1}(\mathbf{K} - \tilde{\mathbf{K}})\|_2\|(\mathbf{K} + \lambda\mathbf{I})^{-1}\mathbf{z}\|_2 \qquad \text{(submultiplicativity)}$$
$$= \frac{1}{\lambda}\operatorname{bias}(\mathbf{K})\left(1 + \|(\tilde{\mathbf{K}} + \lambda\mathbf{I})^{-1}(\mathbf{K} - \tilde{\mathbf{K}})\|_2\right). \tag{35}$$

So we just need to bound $\|(\tilde{\mathbf{K}} + \lambda\mathbf{I})^{-1}(\mathbf{K} - \tilde{\mathbf{K}})\|_2 \le \epsilon$. First note that, by Theorem 3, Corollary 4,

$$\mathbf{K} - \tilde{\mathbf{K}} \preceq \epsilon\lambda\mathbf{I}$$

and since $(\mathbf{K} - \tilde{\mathbf{K}})$ and $\mathbf{I}$ *commute*, it follows that

$$(\mathbf{K} - \tilde{\mathbf{K}})^2 \preceq \epsilon^2\lambda^2\mathbf{I}. \tag{36}$$

Accordingly,

$$\begin{aligned}
\|(\tilde{\mathbf{K}} + \lambda\mathbf{I})^{-1}(\mathbf{K} - \tilde{\mathbf{K}})\|_2^2 &= \|(\tilde{\mathbf{K}} + \lambda\mathbf{I})^{-1}(\mathbf{K} - \tilde{\mathbf{K}})^2(\tilde{\mathbf{K}} + \lambda\mathbf{I})^{-1}\|_2 \\
&\le \epsilon^2\lambda^2 \|(\tilde{\mathbf{K}} + \lambda\mathbf{I})^{-2}\|_2 \\
&\le \epsilon^2\lambda^2 \frac{1}{\lambda^2} = \epsilon^2.
\end{aligned}$$

So $\|(\tilde{\mathbf{K}} + \lambda\mathbf{I})^{-1}(\mathbf{K} - \tilde{\mathbf{K}})\|_2 \le \epsilon$ as desired and plugging into (35) we have shown (34), that $\mathrm{bias}(\tilde{\mathbf{K}}) \le (1 + \epsilon)\mathrm{bias}(\mathbf{K})$. We next show that:

$$\mathrm{variance}(\tilde{\mathbf{K}}) \le \mathrm{variance}(\mathbf{K}), \tag{37}$$

where $\mathrm{variance}(\mathbf{K}) = \sigma^2\,\mathrm{tr}(\mathbf{K}^2(\mathbf{K} + \lambda\mathbf{I})^{-2}) = \sigma^2\sum_{i=1}^{n}\left(\frac{\sigma_i(\mathbf{K})}{\sigma_i(\mathbf{K})+\lambda}\right)^2$. Since $\tilde{\mathbf{K}} \preceq \mathbf{K}$ by Theorem 3, $\sigma_i(\tilde{\mathbf{K}}) \le \sigma_i(\mathbf{K})$ for all $i$. It follows that, for every $i$,

$$\frac{\sigma_i(\tilde{\mathbf{K}})}{\sigma_i(\tilde{\mathbf{K}}) + \lambda} \le \frac{\sigma_i(\mathbf{K})}{\sigma_i(\mathbf{K}) + \lambda}.$$

This in turn implies that

$$\sum_{i=1}^{n}\left(\frac{\sigma_i(\tilde{\mathbf{K}})}{\sigma_i(\tilde{\mathbf{K}}) + \lambda}\right)^2 \le \sum_{i=1}^{n}\left(\frac{\sigma_i(\mathbf{K})}{\sigma_i(\mathbf{K}) + \lambda}\right)^2,$$

which gives (37). Combining (37) and (34) we conclude that, for $\epsilon < 1$,

$$\mathcal{R}(\hat{f}_{\tilde{\mathbf{K}}}) \le (1 + \epsilon)^2\mathcal{R}(\hat{f}_{\mathbf{K}}) \le (1 + 3\epsilon)\mathcal{R}(\hat{f}_{\mathbf{K}}).$$

$\square$

### E.2 Kernel $k$-means

Kernel $k$-means clustering asks us to partition $\mathbf{x}_1, \ldots, \mathbf{x}_n$, into $k$ cluster sets, $\{C_1, \ldots, C_k\}$. Let $\boldsymbol{\mu}_i = \frac{1}{|C_i|}\sum_{\mathbf{x}_j \in C_i} \phi(\mathbf{x}_j)$ be the centroid of the vectors in $C_i$ after mapping to kernel space. The goal is to choose $\{C_1, \ldots, C_k\}$ which minimize the objective:

$$\sum_{i=1}^{k}\sum_{\mathbf{x}_j \in C_i} \|\phi(\mathbf{x}_j) - \boldsymbol{\mu}_i\|_{\mathcal{F}}^2 \tag{38}$$

It is well known that this optimization problem can be rewritten as a *constrained* low-rank approximation problem (see e.g. [BMD09] or [CEM$^+$15]). In particular, for any clustering $C = \{C_1, \ldots, C_k\}$ we can define a rank $k$ orthonormal matrix $\mathbf{C} \in \mathbb{R}^{n \times k}$ called the cluster indicator matrix for $C$. $\mathbf{C}_{i,j} = 1/\sqrt{|C_j|}$ if $\mathbf{x}_i$ is assigned to $C_j$ and $\mathbf{C}_{i,j} = 0$ otherwise. $\mathbf{C}^T\mathbf{C} = \mathbf{I}$, so $\mathbf{C}\mathbf{C}^T$ is a rank $k$ projection matrix. Furthermore, it is not hard to check that:

$$\sum_{i=1}^{k}\sum_{\mathbf{x}_j \in C_i} \|\phi(\mathbf{x}_j) - \boldsymbol{\mu}_i\|_{\mathcal{F}}^2 = \mathrm{tr}\left(\mathbf{K} - \mathbf{C}\mathbf{C}^T\mathbf{K}\mathbf{C}\mathbf{C}^T\right). \tag{39}$$

Informally, if we work with the kernalized data matrix $\boldsymbol{\Phi}$, (39) is equivalent to

$$\|\boldsymbol{\Phi} - \mathbf{C}\mathbf{C}^T\boldsymbol{\Phi}\|_F^2.$$

Regardless, it's clear that solving kernel $k$-means is equivalent to solving:

$$\min_{\mathbf{C} \in \mathcal{S}} \mathrm{tr}\left(\mathbf{K} - \mathbf{C}\mathbf{C}^T\mathbf{K}\mathbf{C}\mathbf{C}^T\right) \tag{40}$$

where $\mathcal{S}$ is the set of all rank $k$ cluster indicator matrices. From this formulation, we easily obtain:

**Theorem 17** (Kernel $k$-means Approximation Bound). *Let $\tilde{\mathbf{K}}$ be computed by RLS-Nyström with $\lambda = \frac{\epsilon}{k} \sum_{i=k+1}^{n} \sigma_i(\mathbf{K})$ and $\delta \in (0, 1/8)$. Let $\tilde{\mathbf{C}}^*$ be the optimal cluster indicator matrix for $\tilde{\mathbf{K}}$ and let $\tilde{\mathbf{C}}$ be an approximately optimal cluster indicator matrix satisfying:*

$$\operatorname{tr}\left(\tilde{\mathbf{K}} - \tilde{\mathbf{C}}\tilde{\mathbf{C}}^T\tilde{\mathbf{K}}\tilde{\mathbf{C}}\tilde{\mathbf{C}}^T\right) \leq (1+\gamma)\operatorname{tr}\left(\tilde{\mathbf{K}} - \tilde{\mathbf{C}}^*\tilde{\mathbf{C}}^{*T}\tilde{\mathbf{K}}\tilde{\mathbf{C}}^*\tilde{\mathbf{C}}^{*T}\right).$$

*Then, if $\mathbf{C}^*$ is the optimal cluster indicator matrix for $\mathbf{K}$:*

$$\operatorname{tr}\left(\mathbf{K} - \tilde{\mathbf{C}}\tilde{\mathbf{C}}^T\mathbf{K}\tilde{\mathbf{C}}\tilde{\mathbf{C}}^T\right) \leq (1+\gamma)(1+\epsilon)\operatorname{tr}\left(\mathbf{K} - \mathbf{C}^*\mathbf{C}^{*T}\mathbf{K}\mathbf{C}^*\mathbf{C}^{*T}\right)$$

*By Theorem 14, Algorithm 3 can compute $\tilde{\mathbf{K}}$ with $O(ns)$ kernel evaluations and $O(ns^2)$ computation time, with $s = O\left(\frac{k}{\epsilon} \log \frac{k}{\delta\epsilon}\right)$.*

In other words, if we find an optimal set of clusters for our approximate kernel matrix, those clusters will provide a $(1+\epsilon)$ approximation to the original kernel $k$-means problem. Furthermore, if we only solve the kernel $k$-means problem approximately on $\tilde{\mathbf{K}}$, i.e. with some approximation factor $(1+\gamma)$, we will do nearly as well on the original problem. This flexibility allows for the use of $k$-means approximation algorithms (since the problem is NP-hard to solve exactly).

*Proof.* The proof is almost immediate from our bounds on RLS-Nyström:

$$\begin{aligned}
\operatorname{tr}\left(\mathbf{K} - \tilde{\mathbf{C}}\tilde{\mathbf{C}}^T\mathbf{K}\tilde{\mathbf{C}}\tilde{\mathbf{C}}^T\right) &\leq \operatorname{tr}\left(\tilde{\mathbf{K}} - \tilde{\mathbf{C}}\tilde{\mathbf{C}}^T\tilde{\mathbf{K}}\tilde{\mathbf{C}}\tilde{\mathbf{C}}^T\right) + c && \text{(Theorem 12)}\\
&\leq (1+\gamma)\operatorname{tr}\left(\tilde{\mathbf{K}} - \tilde{\mathbf{C}}^*\tilde{\mathbf{C}}^{*T}\tilde{\mathbf{K}}\tilde{\mathbf{C}}^*\tilde{\mathbf{C}}^{*T}\right) + (1+\gamma)c && \text{(by assumption)}\\
&\leq (1+\gamma)\operatorname{tr}\left(\tilde{\mathbf{K}} - \mathbf{C}^*\mathbf{C}^{*T}\tilde{\mathbf{K}}\mathbf{C}^*\mathbf{C}^{*T}\right) + (1+\gamma)c && \text{(optimality of } \tilde{\mathbf{C}}^* \text{)}\\
&\leq (1+\gamma)\operatorname{tr}\left(\tilde{\mathbf{K}} - \mathbf{C}^*\mathbf{C}^{*T}\tilde{\mathbf{K}}\mathbf{C}^*\mathbf{C}^{*T}\right) + c && \text{(since } c \geq 0\text{)}\\
&\leq (1+\gamma)(1+\epsilon)\operatorname{tr}\left(\mathbf{K} - \tilde{\mathbf{C}}^*\mathbf{C}^{*T}\mathbf{K}\mathbf{C}^*\mathbf{C}^{*T}\right). && \text{(Theorem 12)}
\end{aligned}$$

$\square$

### E.3 Kernel principal component analysis

We consider the standard formulation of kernel principal component analysis (PCA) presented in [SSM99]. The goal is to find principal components *in the kernel space* $\mathcal{F}$ that capture as much variance in the kernelized data as possible. In particular, if we work informally with the kernelized data matrix $\mathbf{\Phi}$, we want to find a matrix $\mathbf{Z}_k$ containing $k$ orthonormal columns such that:

$$\mathbf{\Phi}\mathbf{\Phi}^T - (\mathbf{\Phi}\mathbf{Z}_k\mathbf{Z}_k^T)(\mathbf{\Phi}\mathbf{Z}_k\mathbf{Z}_k^T)^T$$

is as small as possible. In other words, if we project $\mathbf{\Phi}$'s rows to the $k$ dimensional subspace spanned by $\mathbf{V}_k$'s columns and then recompute our kernel, we want the approximate kernel to be close to the original.

We focus in particular on minimizing PCA error according to the metric:

$$\operatorname{tr}\left(\mathbf{\Phi}\mathbf{\Phi}^T - (\mathbf{\Phi}\mathbf{Z}_k\mathbf{Z}_k^T)(\mathbf{\Phi}\mathbf{Z}_k\mathbf{Z}_k^T)^T\right) = \|\mathbf{\Phi} - \mathbf{\Phi}\mathbf{Z}_k\mathbf{Z}_k^T\|_F^2, \tag{41}$$

which is standard in the literature [Woo14, ANW14]. As with $f$ in kernel ridge regression, to solve this problem we cannot write down $\mathbf{Z}_k$ explicitly for most kernel functions. However, the optimal $\mathbf{Z}_k$ always lies in the column span of $\mathbf{\Phi}^T$, so we can implicitly represent it by constructing a matrix $\mathbf{X} \in \mathbb{R}^{n \times k}$ such that $\mathbf{\Phi}^T\mathbf{X} = \mathbf{Z}_k$. It is then easy to compute the projection of any new data vector onto the span of $\mathbf{Z}_k$ (the typical objective of principal component analysis) since we can multiply by $\mathbf{\Phi}^T\mathbf{X}$ using the kernel function.

By the Eckart-Young theorem the optimal $\mathbf{Z}_k$ contains the top $k$ row principal components of $\mathbf{\Phi}$. Accordingly, if we write the singular value decomposition $\mathbf{\Phi} = \mathbf{U}\mathbf{\Sigma}\mathbf{V}^T$ we want to set $\mathbf{X} = \mathbf{U}_k\mathbf{\Sigma}_k^{-1}$,

which can be computed from the SVD of $\mathbf{K} = \mathbf{U}\boldsymbol{\Sigma}^2\mathbf{U}^T$. $\mathbf{Z}_k$ will equal $\mathbf{V}_k$ and (41) reduces to:

$$\mathrm{tr}(\mathbf{K} - \boldsymbol{\Phi}\mathbf{V}_k\mathbf{V}_k^T\boldsymbol{\Phi}) = \mathrm{tr}(\mathbf{K} - \mathbf{V}_k\mathbf{V}_k^T\mathbf{K}) \qquad \text{(cyclic property)}$$

$$= \sum_{i=k+1}^{n} \sigma_i(\mathbf{K}). \tag{42}$$

**Theorem 18** (Kernel PCA Approximation Bound). *Let $\tilde{\mathbf{K}}$ be computed by RLS-Nyström with $\lambda = \frac{\epsilon}{k}\sum_{i=k+1}^{n}\sigma_i(\mathbf{K})$ and $\delta \in (0, 1/8)$. From $\tilde{\mathbf{K}}$ we can compute a matrix $\mathbf{X} \in \mathbb{R}^{s \times k}$ such that if we set $\mathbf{Z} = \boldsymbol{\Phi}^T\mathbf{S}\mathbf{X}$, with probability $1 - \delta$:*

$$\|\boldsymbol{\Phi} - \boldsymbol{\Phi}\mathbf{Z}\mathbf{Z}^T\|_F^2 \le (1 + 2\epsilon)\|\boldsymbol{\Phi} - \boldsymbol{\Phi}\mathbf{V}_k\mathbf{V}_k^T\|_F^2 = (1 + 2\epsilon)\sum_{i=k+1}^{n}\sigma_i(\mathbf{K}).$$

*By Theorem 14, Algorithm 3 can compute $\tilde{\mathbf{K}}$ with $O(ns)$ kernel evaluations and $O(ns^2)$ computation time, with $s = O\left(\frac{k}{\epsilon}\log\frac{k}{\delta\epsilon}\right)$.*

Note that $\mathbf{S}$ is the sampling matrix used to construct $\tilde{\mathbf{K}}$. $\mathbf{Z} = \boldsymbol{\Phi}^T\mathbf{S}\mathbf{X}$ can be applied to vectors (in order to project onto the approximate low-rank subspace) using only $s$ kernel evaluations.

*Proof.* Re-parameterizing $\mathbf{Z}_k = \boldsymbol{\Phi}^T\mathbf{Y}$, we see that minimizing (41) is equivalent to minimizing

$$\mathrm{tr}(\mathbf{K} - \mathbf{K}\mathbf{Y}\mathbf{Y}^T\mathbf{K})$$

over $\mathbf{Y} \in \mathbb{R}^{n \times k}$ such that $(\boldsymbol{\Phi}^T\mathbf{Y})^T\boldsymbol{\Phi}^T\mathbf{Y} = \mathbf{Y}^T\mathbf{K}\mathbf{Y} = \mathbf{I}$. Then we re-parameterize again by writing $\mathbf{Y} = \mathbf{K}^{-1/2}\mathbf{W}$ where $\mathbf{W}$ is an $n \times k$ matrix with orthonormal columns. Using linearity and cyclic property of the trace, we can write:

$$\mathrm{tr}(\mathbf{K} - \mathbf{K}\mathbf{Y}\mathbf{Y}^T\mathbf{K}) = \mathrm{tr}(\mathbf{K}) - \mathrm{tr}(\mathbf{Y}^T\mathbf{K}\mathbf{K}\mathbf{Y}) = \mathrm{tr}(\mathbf{K}) - \mathrm{tr}(\mathbf{W}^T\mathbf{K}\mathbf{W}) = \mathrm{tr}(\mathbf{K}) - \mathrm{tr}(\mathbf{W}\mathbf{W}^T\mathbf{K}\mathbf{W}\mathbf{W}^T).$$

So, we have reduced our problem to a low-rank approximation problem that looks exactly like the $k$-means problem from Section E.2, except without constraints.

Accordingly, following the same argument as Theorem 17, if we find $\tilde{\mathbf{W}}$ minimizing:

$$\mathrm{tr}(\tilde{\mathbf{K}}) - \mathrm{tr}(\tilde{\mathbf{W}}\tilde{\mathbf{W}}^T\tilde{\mathbf{K}}\tilde{\mathbf{W}}\tilde{\mathbf{W}}^T),$$

then:

$$\mathrm{tr}(\mathbf{K}) - \mathrm{tr}(\tilde{\mathbf{W}}\tilde{\mathbf{W}}^T\mathbf{K}\tilde{\mathbf{W}}\tilde{\mathbf{W}}^T) \le (1 + \epsilon)\left[\min_{\mathbf{W}}\mathrm{tr}(\mathbf{K}) - \mathrm{tr}(\mathbf{W}\mathbf{W}^T\mathbf{K}\mathbf{W}\mathbf{W}^T)\right] = (1 + \epsilon)\sum_{i=k+1}^{n}\sigma_i(\mathbf{K}).$$

$\tilde{\mathbf{W}}$ can be taken to equal the top $k$ eigenvectors of $\tilde{\mathbf{K}}$, which can be found in $O(n \cdot s^2)$ time.

However, we are not quite done. Thanks to our re-parameterization this bound guarantees that $\boldsymbol{\Phi}^T\mathbf{K}^{-1/2}\tilde{\mathbf{W}}$ is a good set of approximate kernel principal components for $\boldsymbol{\Phi}$. Unfortunately, $\boldsymbol{\Phi}^T\mathbf{K}^{-1/2}\tilde{\mathbf{W}}$ cannot be represented efficiently (it requires computing $\mathbf{K}^{-1/2}$) and projecting new vectors to $\boldsymbol{\Phi}^T\mathbf{K}^{-1/2}\tilde{\mathbf{W}}$ would require $n$ kernel evaluations to multiply by $\boldsymbol{\Phi}^T$.

Instead, recalling the definition of $\mathbf{P_S} = \boldsymbol{\Phi}^T\mathbf{S}(\mathbf{S}^T\mathbf{K}^T\mathbf{S})^+\mathbf{S}^T\boldsymbol{\Phi}$ from Section 2.1, we suggest using the approximate principal components:

$$\mathbf{P_S}\boldsymbol{\Phi}^T\tilde{\mathbf{K}}^{-1/2}\tilde{\mathbf{W}}.$$

Clearly $\mathbf{P_S}\boldsymbol{\Phi}^T\tilde{\mathbf{K}}^{-1/2}\tilde{\mathbf{W}}$ is orthonormal because:

$$(\mathbf{P_S}\boldsymbol{\Phi}^T\tilde{\mathbf{K}}^{-1/2}\tilde{\mathbf{W}})^T\mathbf{P_S}\boldsymbol{\Phi}^T\tilde{\mathbf{K}}^{-1/2}\tilde{\mathbf{W}} = \tilde{\mathbf{W}}^T\tilde{\mathbf{K}}^{-1/2}\boldsymbol{\Phi}^T\mathbf{P_S}\boldsymbol{\Phi}\tilde{\mathbf{K}}^{-1/2}\tilde{\mathbf{W}}$$

$$= \tilde{\mathbf{W}}^T\mathbf{I}\tilde{\mathbf{W}} = \mathbf{I}.$$

We will argue that it is offers nearly as a good of a solution as $\mathbf{\Phi}^T \mathbf{K}^{-1/2}\tilde{\mathbf{W}}$. Specifically, substituting into (41) gives a value of:

$$\mathrm{tr}(\mathbf{K} - \mathbf{\Phi}\mathbf{P_S}\mathbf{\Phi}^T\tilde{\mathbf{K}}^{-1/2}\tilde{\mathbf{W}}\tilde{\mathbf{W}}^T\tilde{\mathbf{K}}^{-1/2}\mathbf{\Phi}\mathbf{P_S}\mathbf{\Phi}^T) = \mathrm{tr}(\mathbf{K}) - \mathrm{tr}(\tilde{\mathbf{W}}\tilde{\mathbf{W}}^T\tilde{\mathbf{K}}^{-1/2}\mathbf{\Phi}\mathbf{P_S}\mathbf{\Phi}^T\mathbf{\Phi}\mathbf{P_S}\mathbf{\Phi}^T\tilde{\mathbf{K}}^{-1/2})$$
$$= \mathrm{tr}(\mathbf{K}) - \mathrm{tr}(\tilde{\mathbf{W}}\tilde{\mathbf{W}}^T\tilde{\mathbf{K}}^{-1/2}\tilde{\mathbf{K}}^2\tilde{\mathbf{K}}^{-1/2})$$
$$= \mathrm{tr}(\mathbf{K}) - \mathrm{tr}(\tilde{\mathbf{W}}\tilde{\mathbf{W}}^T\tilde{\mathbf{K}}).$$

Compare this to the value obtained from $\mathbf{\Phi}^T\mathbf{K}^{-1/2}\tilde{\mathbf{W}}$:

$$\left[\mathrm{tr}(\mathbf{K}) - \mathrm{tr}(\tilde{\mathbf{W}}\tilde{\mathbf{W}}^T\mathbf{K}\tilde{\mathbf{W}}\tilde{\mathbf{W}}^T)\right] - \left[\mathrm{tr}(\mathbf{K}) - \mathrm{tr}(\tilde{\mathbf{W}}\tilde{\mathbf{W}}^T\tilde{\mathbf{K}}\tilde{\mathbf{W}}\tilde{\mathbf{W}}^T)\right]$$

$$= \mathrm{tr}\left(\tilde{\mathbf{W}}\tilde{\mathbf{W}}^T(\mathbf{K} - \tilde{\mathbf{K}})\right) = \mathrm{tr}\left(\tilde{\mathbf{W}}^T(\mathbf{K} - \tilde{\mathbf{K}})\tilde{\mathbf{W}}\right) = \sum_{i=1}^{k} \tilde{\mathbf{w}}_i^T(\mathbf{K} - \tilde{\mathbf{K}})\tilde{\mathbf{w}}_i \leq k\frac{\epsilon}{k}\sum_{i=k+1}^{n}\sigma_i(\mathbf{K}).$$

$$(43)$$

The last step follows from Theorem 3 which guarantees that $(\mathbf{K} - \tilde{\mathbf{K}}) \preceq \epsilon\lambda\mathbf{I}$. Recall that we set $\lambda = \frac{\epsilon}{k}\sum_{i=k+1}^{n}\sigma_i(\mathbf{K})$ and each column $\tilde{\mathbf{w}}_i$ of $\tilde{\mathbf{W}}$ has unit norm.

We conclude that the cost obtained by $\mathbf{P_S}\mathbf{\Phi}^T\tilde{\mathbf{K}}^{-1/2}\tilde{\mathbf{W}}$ is bounded by:

$$\mathrm{tr}(\mathbf{K} - \mathbf{\Phi}\mathbf{P_S}\mathbf{\Phi}^T\tilde{\mathbf{K}}^{-1/2}\tilde{\mathbf{W}}\tilde{\mathbf{W}}^T\tilde{\mathbf{K}}^{-1/2}\mathbf{\Phi}\mathbf{P_S}\mathbf{\Phi}^T) \leq \mathrm{tr}(\mathbf{K}) - \mathrm{tr}(\tilde{\mathbf{W}}\tilde{\mathbf{W}}^T\mathbf{K}\tilde{\mathbf{W}}\tilde{\mathbf{W}}^T) + \epsilon\sum_{i=k+1}^{n}\sigma_i(\mathbf{K})$$

$$\leq (1 + 2\epsilon)\sum_{i=k+1}^{n}\sigma_i(\mathbf{K}).$$

This gives the result. Notice that $\mathbf{P_S}\mathbf{\Phi}^T\tilde{\mathbf{K}}^{-1/2}\tilde{\mathbf{W}} = \mathbf{\Phi}^T\mathbf{S}(\mathbf{S}^T\mathbf{K}^T\mathbf{S})^+\mathbf{S}^T\mathbf{\Phi}\mathbf{\Phi}^T\tilde{\mathbf{K}}^{-1/2}\tilde{\mathbf{W}}$ so, if we set:

$$\mathbf{X} = (\mathbf{S}^T\mathbf{K}^T\mathbf{S})^+\mathbf{S}^T\tilde{\mathbf{K}}^{1/2}\tilde{\mathbf{W}},$$

our solution can be represented as $\mathbf{Z} = \mathbf{\Phi}^T\mathbf{S}\mathbf{X}$ as desired. $\qquad\square$

### E.4  Kernel canonical correlation analysis

We briefly discuss a final application to canonical correlation analysis (CCA) that follows from applying our spectral approximation guarantee of Theorem 3 to recent work in [Wan16].

Consider $n$ pairs of input points $(\mathbf{x}_1, \mathbf{y}_1), ..., (\mathbf{x}_n, \mathbf{y}_n) \in (\mathcal{X}, \mathcal{Y})$ along with two positive semidefinite kernels, $K_x : \mathcal{X} \times \mathcal{X} \rightarrow \mathbb{R}$ and $K_y : \mathcal{Y} \times \mathcal{Y} \rightarrow \mathbb{R}$. Let $\mathcal{F}_x$ and $\mathcal{F}_y$ and $\phi_x : \mathcal{X} \rightarrow \mathcal{F}_x$ and $\phi_y : \mathcal{Y} \rightarrow \mathcal{F}_y$ be the Hilbert spaces and feature maps associated with these kernels. Let $\mathbf{\Phi}_x$ and $\mathbf{\Phi}_y$ denote the kernelized $\mathcal{X}$ and $\mathcal{Y}$ inputs respectively and $\mathbf{K}_x$ and $\mathbf{K}_y$ denote the associated kernel matrices.

We consider standard regularized kernel CCA, following the presentation in [Wan16]. The goal is to compute coefficient vectors $\boldsymbol{\alpha}^x$ and $\boldsymbol{\alpha}^y$ such that $\mathbf{f}_x^* = \sum_{i=1}^{n}\boldsymbol{\alpha}_i^x\phi_x(\mathbf{x}_i)$ and $\mathbf{f}_y^* = \sum_{i=1}^{n}\boldsymbol{\alpha}_i^y\phi_y(\mathbf{y}_i)$ satisfy:

$$(\mathbf{f}_x^*, \mathbf{f}_y^*) = \operatorname*{arg\,max}_{\mathbf{f}_x \in \mathcal{F}_x, \mathbf{f}_y \in \mathcal{F}_y} \mathbf{f}_x^T\mathbf{\Phi}_x^T\mathbf{\Phi}_y\mathbf{f}_y^*$$

subject to

$$\mathbf{f}_x^T\mathbf{\Phi}_x^T\mathbf{\Phi}_x\mathbf{f}_x + \lambda_x\|\mathbf{f}_x\|_{\mathcal{F}_x}^2 = 1$$
$$\mathbf{f}_y^T\mathbf{\Phi}_y^T\mathbf{\Phi}_y\mathbf{f}_y + \lambda_y\|\mathbf{f}_y\|_{\mathcal{F}_y}^2 = 1$$

In [Wan16], the kernelized points are centered to their means. For simplicity we ignore centering, but note that [Wan16] shows how bounds for the uncentered problem carry over to the centered one.

It can be shown that $\boldsymbol{\alpha}^x = (\mathbf{K}_x + \lambda_x\mathbf{I})^{-1}\boldsymbol{\beta}^x$ and $\boldsymbol{\alpha}^y = (\mathbf{K}_y + \lambda_y\mathbf{I})^{-1}\boldsymbol{\beta}^y$ where $\boldsymbol{\beta}^x$ and $\boldsymbol{\beta}^y$ are the top left and right singular vectors respectively of

$$\mathbf{T} = (\mathbf{K}_x + \lambda_x\mathbf{I})^{-1}\mathbf{K}_x\mathbf{K}_y(\mathbf{K}_y + \lambda_y\mathbf{I})^{-1}.$$

The optimum value of the above program will be equal to $\sigma_1(\mathbf{T})$.

[Wan16] shows that if $\tilde{\mathbf{K}}_x$ and $\tilde{\mathbf{K}}_y$ satisfy:

$$\tilde{\mathbf{K}}_x \preceq \mathbf{K}_x \preceq \tilde{\mathbf{K}}_x + \epsilon\lambda_x\mathbf{I}$$
$$\tilde{\mathbf{K}}_y \preceq \mathbf{K}_y \preceq \tilde{\mathbf{K}}_y + \epsilon\lambda_x\mathbf{I}$$

then if $\tilde{\boldsymbol{\alpha}}^x$ and $\tilde{\boldsymbol{\alpha}}^y$ are computed using these approximations, the achieved objective function value will be within $\epsilon$ of optimal (see their Lemma 1 and Theorem 1). So we have:

**Theorem 19** (Kernel CCA Approximation Bound). *Suppose $\tilde{\mathbf{K}}_x$ and $\tilde{\mathbf{K}}_y$ are computed by RLS-Nyström with approximation parameters $\epsilon\lambda_x$ and $\epsilon\lambda_y$ and failure probability $\delta \in (0, 1/8)$. If we solve for $\tilde{\boldsymbol{\alpha}}^x$ and $\tilde{\boldsymbol{\alpha}}^y$, the approximate canonical correlation will be within an additive $\epsilon$ of the true canonical correlation $\sigma_1(\mathbf{T})$.*

*By Theorem 7, Algorithm 2 can compute $\tilde{\mathbf{K}}_x$ and $\tilde{\mathbf{K}}_y$ with $O(ns_x + ns_y)$ kernel evaluations and $O(ns_x^2 + ns_y^2)$ computation time, with $s_x = O\left(\frac{d_{\text{eff}}^{\lambda_x}}{\epsilon} \log \frac{d_{\text{eff}}^{\lambda_x}}{\delta\epsilon}\right)$ and $s_y = O\left(\frac{d_{\text{eff}}^{\lambda_y}}{\epsilon} \log \frac{d_{\text{eff}}^{\lambda_y}}{\delta\epsilon}\right)$.*

# F  Additional proofs

## F.1  Ridge leverage score approximation via uniform sampling

**Lemma 5.** *For any $\mathbf{B} \in \mathbb{R}^{n \times n}$ with $\mathbf{BB}^T = \mathbf{K}$ and $\mathbf{S} \in \mathbb{R}^{n \times s}$ chosen by sampling each data point independently with probability $1/2$, let*

$$\tilde{l}_i^\lambda = \mathbf{b}_i^T(\mathbf{B}^T\mathbf{SS}^T\mathbf{B} + \lambda\mathbf{I})^{-1}\mathbf{b}_i \tag{44}$$

*and $p_i = \min\{1, 16\tilde{l}_i^\lambda \log(\sum_i \tilde{l}_i^\lambda/\delta)\}$ for any $\delta \in (0, 1/8)$. Then with probability at least $1 - \delta$:*

1. *$\tilde{l}_i^\lambda \geq l_i^\lambda$ for all $i$.*

2. *$\sum_i p_i \leq 64 \sum_i l_i^\lambda \log(\sum_i l_i^\lambda/\delta)$.*

*Proof.* The first bound follows trivially since $\mathbf{B}^T\mathbf{SS}^T\mathbf{B} \preceq \mathbf{B}^T\mathbf{B}$ so:

$$\tilde{l}_i^\lambda = \mathbf{b}_i^T(\mathbf{B}^T\mathbf{SS}^T\mathbf{B} + \lambda\mathbf{I})^{-1}\mathbf{b}_i \geq \mathbf{b}_i^T(\mathbf{B}^T\mathbf{B} + \lambda\mathbf{I})^{-1}\mathbf{b}_i = l_i^\lambda.$$

The challenge is showing the second bound. The key observation is that there exists a diagonal reweighting matrix $\mathbf{W} \in \mathbb{R}^{n \times n}$, $\mathbf{0} \preceq \mathbf{W} \preceq \mathbf{I}$ such that for all $i$, $l_i^\lambda(\mathbf{WKW}) \leq \alpha$ where $\alpha \stackrel{\text{def}}{=} \frac{1}{2} \cdot \frac{1}{16 \log(\sum l_i^\lambda/\delta)}$. This bound ensures that uniformly sampling rows with probability $1/2$ from the *reweighted kernel* $\mathbf{WKW}$ is a valid ridge leverage score sampling. Additionally, $|\{i : \mathbf{W}_{i,i} < 1\}| \leq 32 \log(\sum l_i^\lambda/\delta) \cdot \sum l_i^\lambda$. That is, we do not need to reweight too many columns to achieve the ridge leverage score upper bound.

Although $\mathbf{W}$ is never actually computed, its existence can be proved algorithmically: we can construct a valid $\mathbf{W}$ by iteratively considering any $i$ with $l_i^\lambda(\mathbf{WKW}) \geq \alpha$. Since $\lambda > 0$, it is always possible to decrease the ridge leverage score to exactly $\alpha$ by decreasing $\mathbf{W}_{i,i}$ sufficiently.

It is clear from the interpretation of Definition 1 given in (4) that decreasing $\mathbf{W}_{i,i}$, which corresponds to decreasing the weight of one row of $\mathbf{B}$, will only increase the ridge leverage scores of other rows. So, any reweighted row will always maintain leverage score $\geq \alpha$ as other rows are reweighted. Theorem 2 of [CLM+15] demonstrates rigorously that the leverage scores of these reweighted rows in fact converge to $\alpha$. Furthermore, since $\mathbf{W} \preceq \mathbf{I}$, it is not hard to show (see Lemma 20):

$$\sum_i l_i^\lambda(\mathbf{WKW}) \leq \sum_i l_i^\lambda(\mathbf{K}) \stackrel{\text{def}}{=} \sum_i l_i^\lambda.$$

Thus, since each reweighted row has $l_i^\lambda(\mathbf{WKW}) \geq \alpha$, $\alpha \cdot |\{i : \mathbf{W}_{i,i} < 1\}| \leq \sum_i l_i^\lambda$ and so:

$$|\{i : \mathbf{W}_{i,i} < 1\}| \leq \frac{1}{\alpha} \sum_i l_i^\lambda = 32 \log\left(\sum l_i^\lambda/\delta\right) \cdot \sum l_i^\lambda.$$

We can now bound $\sum_i p_i$. For any $i$ that is reweighted by $\mathbf{W}$ we just trivially bound $p_i \le 1$. Since $l_i^\lambda(\mathbf{WKW}) \le \frac{1}{2} \cdot \frac{1}{16 \log(\sum l_i^\lambda/\delta)}$ for all $i$, and since $\mathbf{S}$ samples each $i$ with probability $1/2$, by the matrix Bernstein bound of Lemma 9, with probability $1 - \delta/2$:

$$\frac{1}{2}(\mathbf{B}^T\mathbf{W}^2\mathbf{B} + \lambda\mathbf{I}) \preceq (\mathbf{B}^T\mathbf{WSS}^T\mathbf{WB} + \lambda\mathbf{I}) \preceq \frac{3}{2}(\mathbf{B}^T\mathbf{W}^2\mathbf{B} + \lambda\mathbf{I}).$$

Hence:

$$\tilde{l}_i^\lambda = \mathbf{b}_i^T(\mathbf{B}^T\mathbf{SS}^T\mathbf{B} + \lambda\mathbf{I})^{-1}\mathbf{b}_i \le \mathbf{b}_i^T(\mathbf{B}^T\mathbf{WSS}^T\mathbf{WB} + \lambda\mathbf{I})^{-1}\mathbf{b}_i$$
$$\le 2\mathbf{b}_i^T(\mathbf{B}^T\mathbf{W}^2\mathbf{B} + \lambda\mathbf{I})^{-1}\mathbf{b}_i$$
$$= 2l_i^\lambda(\mathbf{WBB}^T\mathbf{W}) = 2l_i^\lambda(\mathbf{WKW}).$$

Again using that $\mathbf{W} \preceq \mathbf{I}$ and Lemma 20, $\sum_{\{i:\mathbf{W}_{i,i}=1\}} \tilde{l}_i^\lambda \le 2\sum_i l_i^\lambda$. Overall:

$$\sum_i p_i = \sum_{\{i:\mathbf{W}_{i,i}<1\}} p_i + \sum_{\{i:\mathbf{W}_{i,i}=1\}} p_i$$
$$\le |\{i : \mathbf{W}_{i,i} < 1\}| + 32 \log\left(\sum l_i^\lambda/\delta\right) \cdot \sum_i l_i^\lambda$$
$$= 64 \log\left(\sum l_i^\lambda/\delta\right) \cdot \sum_i l_i^\lambda.$$

$\square$

### F.2 Formula for ridge leverage score computation

**Lemma 6.** *For any sampling matrix $\mathbf{S} \in \mathbb{R}^{n\times s}$, and any $\lambda > 0$:*
$$\tilde{l}_i^\lambda \overset{\text{def}}{=} \mathbf{b}_i^T(\mathbf{B}^T\mathbf{SS}^T\mathbf{B} + \lambda\mathbf{I})^{-1}\mathbf{b}_i = \frac{1}{\lambda}\left(\mathbf{K} - \mathbf{KS}\left(\mathbf{S}^T\mathbf{KS} + \lambda\mathbf{I}\right)^{-1}\mathbf{S}^T\mathbf{K}\right)_{i,i}.$$

*It follows that we can compute $\tilde{l}_i^\lambda$ for all $i$ in $O(ns^2)$ time using just $O(ns)$ kernel evaluations.*

*Proof.* Using the SVD write $\mathbf{S}^T\mathbf{B} = \bar{\mathbf{U}}\bar{\mathbf{\Sigma}}\bar{\mathbf{V}}^T$. $\bar{\mathbf{V}} \in \mathbb{R}^{n\times s}$ forms an orthonormal basis for the row span of $\mathbf{S}^T\mathbf{B}$. Let $\bar{\mathbf{V}}_\perp$ be span for the nullspace of $\mathbf{S}^T\mathbf{B}$. Then we can rewrite $\tilde{l}_i^\lambda$ as:

$$\tilde{l}_i^\lambda = \mathbf{b}_i^T\left(\mathbf{B}^T\mathbf{SS}^T\mathbf{B} + \lambda\mathbf{I}\right)^{-1}\mathbf{b}_i = \mathbf{b}_i^T\left[\bar{\mathbf{V}}, \bar{\mathbf{V}}_\perp\right](\bar{\mathbf{\Sigma}}^2 + \lambda\mathbf{I})^{-1}\left[\bar{\mathbf{V}}, \bar{\mathbf{V}}_\perp\right]^T\mathbf{b}_i.$$

Here we're abusing notation a bit by letting $\bar{\mathbf{\Sigma}}$ represent an $n \times n$ diagonal matrix whose first $s$ entries are the singular values of $\mathbf{S}^T\mathbf{B}$ and whose remaining entries are all equal to 0. Now:

$$\tilde{l}_i^\lambda = \mathbf{b}_i^T\left[\bar{\mathbf{V}}, \bar{\mathbf{V}}_\perp\right](\bar{\mathbf{\Sigma}}^2 + \lambda\mathbf{I})^{-1}\left[\bar{\mathbf{V}}, \bar{\mathbf{V}}_\perp\right]^T\mathbf{b}_i = \frac{1}{\lambda}\mathbf{b}_i^T\bar{\mathbf{V}}_\perp^T\bar{\mathbf{V}}_\perp\mathbf{b}_i + \mathbf{b}_i^T\bar{\mathbf{V}}(\bar{\mathbf{\Sigma}}^2 + \lambda\mathbf{I})^{-1}\bar{\mathbf{V}}^T\mathbf{b}_i^T.$$
(45)

Focusing on the second term of (45),

$$\mathbf{b}_i^T\bar{\mathbf{V}}(\bar{\mathbf{\Sigma}}^2 + \lambda\mathbf{I})^{-1}\bar{\mathbf{V}}^T\mathbf{b}_i = \mathbf{b}_i^T\bar{\mathbf{V}}\frac{1}{\lambda}\left(\mathbf{I} - \bar{\mathbf{\Sigma}}^2(\bar{\mathbf{\Sigma}}^2 + \lambda\mathbf{I})^{-1}\right)\bar{\mathbf{V}}^T\mathbf{b}_i$$
$$= \frac{1}{\lambda}\mathbf{b}_i^T\bar{\mathbf{V}}\bar{\mathbf{V}}^T\mathbf{b}_i - \frac{1}{\lambda}\mathbf{b}_i^T\bar{\mathbf{V}}\left(\bar{\mathbf{\Sigma}}^2(\bar{\mathbf{\Sigma}}^2 + \lambda\mathbf{I})^{-1}\right)\bar{\mathbf{V}}^T\mathbf{b}_i.$$ (46)

Focusing on the second term of (46),

$$\mathbf{b}_i^T\bar{\mathbf{V}}\left(\bar{\mathbf{\Sigma}}^2(\bar{\mathbf{\Sigma}}^2 + \lambda\mathbf{I})^{-1}\right)\bar{\mathbf{V}}^T\mathbf{b}_i = \mathbf{b}_i^T\bar{\mathbf{V}}\bar{\mathbf{\Sigma}}\bar{\mathbf{U}}^T\bar{\mathbf{U}}(\bar{\mathbf{\Sigma}}^2 + \lambda\mathbf{I})^{-1}\bar{\mathbf{U}}^T\bar{\mathbf{U}}\bar{\mathbf{\Sigma}}\bar{\mathbf{V}}^T\mathbf{b}_i^T$$
$$= \mathbf{b}_i^T\mathbf{B}^T\mathbf{S}(\mathbf{S}^T\mathbf{KS} + \lambda\mathbf{I})^{-1}\mathbf{S}^T\mathbf{Bb}_i.$$

Substituting back into (46) and then (45), we conclude that:

$$\tilde{l}_i^\lambda = \frac{1}{\lambda}\mathbf{b}_i^T\bar{\mathbf{V}}_\perp^T\bar{\mathbf{V}}_\perp\mathbf{b}_i + \frac{1}{\lambda}\mathbf{b}_i^T\bar{\mathbf{V}}\bar{\mathbf{V}}^T\mathbf{b}_i - \frac{1}{\lambda}\mathbf{b}_i^T\mathbf{B}^T\mathbf{S}(\mathbf{S}^T\mathbf{KS} + \lambda\mathbf{I})^{-1}\mathbf{S}^T\mathbf{Bb}_i$$
$$= \frac{1}{\lambda}\mathbf{b}_i^T\mathbf{b}_i - \frac{1}{\lambda}\mathbf{b}_i^T\mathbf{B}^T\mathbf{S}(\mathbf{S}^T\mathbf{KS} + \lambda\mathbf{I})^{-1}\mathbf{S}^T\mathbf{Bb}_i$$
$$= \frac{1}{\lambda}\mathbf{K}_{i,i} - \frac{1}{\lambda}\left(\mathbf{KS}\left(\mathbf{S}^T\mathbf{KS} + \lambda\mathbf{I}\right)^{-1}\mathbf{S}^T\mathbf{K}\right)_{i,i}.$$

We can compute $(\mathbf{S}^T\mathbf{KS} + \lambda\mathbf{I})^{-1}$ in $O(s^3) \leq O(ns^2)$ time and $O(s^2) \leq O(ns)$ kernel evaluations. Given this inverse, computing the diagonal entries of $\mathbf{KS}\left(\mathbf{S}^T\mathbf{KS} + \lambda\mathbf{I}\right)^{-1}\mathbf{S}^T\mathbf{K}$ requires just $O(ns)$ kernel evaluations to form $\mathbf{KS}$ and $O(ns^2)$ time to perform the necessary multiplications. Finally, computing the diagonal entries of $\mathbf{K}$ requires $n$ additional kernel evaluations. $\qquad\square$

### F.3 Effective dimension bound

**Lemma 20.** *For any* $\mathbf{W} \in \mathbb{R}^{n \times p}$ *with* $\mathbf{WW}^T \preceq \mathbf{I}$,

$$\sum_{i=1}^{n} l_i^\lambda(\mathbf{W}^T\mathbf{KW}) \leq \sum_{i=1}^{n} l_i^\lambda(\mathbf{K}),$$

*or equivalently, by Fact 2,*

$$d_{eff}^\lambda(\mathbf{W}^T\mathbf{KW}) \leq d_{eff}^\lambda(\mathbf{K}).$$

*Proof.* By Definition 1, $l_i^\lambda = \left(\mathbf{K}(\mathbf{K} + \lambda\mathbf{I})^{-1}\right)_{i,i}$ so

$$\sum_{i=1}^{n} l_i^\lambda(\mathbf{K}) = \operatorname{tr}\left(\mathbf{K}(\mathbf{K} + \lambda\mathbf{I})^{-1}\right) = \sum_{i=1}^{n} \frac{\sigma_i(\mathbf{K})}{\sigma_i(\mathbf{K}) + \lambda}.$$

Take any matrix $\mathbf{B} \in \mathbb{R}^{n \times n}$ such that $\mathbf{BB}^T = \mathbf{K}$. Note that for any matrix $\mathbf{Y}$, $\sigma_i(\mathbf{YY}^T) = \sigma_i(\mathbf{Y}^T\mathbf{Y})$ for any non-zero singular values. Accordingly,

$$\sigma_i(\mathbf{W}^T\mathbf{KW}) = \sigma_i(\mathbf{W}^T\mathbf{BB}^T\mathbf{W}) = \sigma_i(\mathbf{B}^T\mathbf{WW}^T\mathbf{B}) \leq \sigma_i(\mathbf{B}^T\mathbf{B}) = \sigma_i(\mathbf{BB}^T) = \sigma_i(\mathbf{K})$$

The $\leq$ step follows from $\mathbf{WW}^T \preceq \mathbf{I}$ so $\mathbf{B}^T\mathbf{WW}^T\mathbf{B} \preceq \mathbf{B}^T\mathbf{B}$. We thus have:

$$\sum_{i=1}^{n} l_i^\lambda(\mathbf{W}^T\mathbf{KW}) = \sum_{i=1}^{p} \frac{\sigma_i(\mathbf{W}^T\mathbf{KW})}{\sigma_i(\mathbf{W}^T\mathbf{KW}) + \lambda} \leq \sum_{i=1}^{n} \frac{\sigma_i(\mathbf{K})}{\sigma_i(\mathbf{K}) + \lambda} = \sum_{i=1}^{n} l_i^\lambda(\mathbf{K}),$$

giving the lemma. $\qquad\square$

## G  Additional empirical results

### G.1  Accelerated recursive method

While Recursive RLS-Nyström typically outperforms classic Nyström, on datasets with relatively uniform ridge leverage scores, such as `YearPredictionMSD`, it only narrowly beats uniform sampling in terms accuracy. As a result it incurs a higher runtime cost since it is slower per sample (see Figure 3).

To combat this issue we implement a simple heuristic modification of our algorithm. We note that the final cost of computing the Nyström factors $\mathbf{KS}$ and $(\mathbf{S}^T\mathbf{KS})^+$ is $O(ns + s^3)$ for both methods. Recursive RLS-Nyström is only slower because computing leverage scores at intermediate levels of recursion takes $O(ns^2)$ time (Step 9, Algorithm 3) . This cost can be improved by simply adjusting the regularization $\lambda$ to restrict the sample size on each recursive call to be $< s$. Specifically, we can balance runtimes by taking $\approx \sqrt{(ns + s^3)/n}$ samples on lower levels.

Doing so improves our runtime, bringing the per sample cost down to approximately that of random Fourier features and uniform Nyström (Figure 5a) while nearly maintaining the same approximation quality.

For datasets such as `Covertype` in which Recursive RLS-Nyström performs significantly better than uniform sampling, so does the accelerated method (see Figure 5b). However, the performance of the accelerated method does not degrade when leverage scores are relatively uniform – it still offers the best runtime to approximation quality tradeoff (Figure 5c).

We note further runtime optimizations may be possible. Subsequent work extends fast ridge leverage score methods to distributed and streaming environments [CLV17]. Empirical evaluation of these techniques could lead to even more scalable, high accuracy Nyström methods.

(a) Runtimes for `Covertype`.

(b) Errors for `Covertype`.

(c) Runtime/error tradeoff for `YearPredictionMSD`.

Figure 5: Our accelerated Recursive RLS-Nyström, which undersamples at intermediate recursive calls, nearly matches the *per sample runtime* of random Fourier features and uniform Nyström while still providing approximation nearly as good as the standard Recursive RLS-Nyström. For datasets like `YearPredictionMSD` with relatively uniform kernel leverage scores, the accelerated version offers the best runtime vs. approximation tradeoff. All results are averaged over 10 trials.

(a) `Covertype`

(b) `YearPredictionMSD`

Figure 6: Performance of kernel approximation methods for classification and clustering. For `Covertype`, classification error is measured in separating Class 2 from the remaining classes. For `YearPredictionMSD`, RMSE is for the unnormalized output. Regularization and kernel parameters are obtained via cross validation on training data. Test results are averaged over 10 trials with a fixed test set, as all three algorithms are randomized.

## G.2   Performance of Recursive RLS-Nyström for learning tasks

We verify the usefulness of our kernel approximations in downstream learning tasks. We focus on `Covertype` and `YearPredictionMSD`, which each have approximately $n = 500,000$ data points. While full kernel methods do not scale in this regime, Recursive RLS-Nyström does since its runtime depends linearly on $n$. For example, on `YearPredictionMSD` the method requires 307 sec. (averaged over 5 trials) to build a $2,000$ landmark Nyström approximation for $463,716$ training points. Ridge regression using the approximate kernel then requires 208 sec. for a total of 515 sec. In comparison, the fastest method, random Fourier features, required 43 sec. to build a rank $2,000$ kernel approximation and 222 sec. for regression, for a total time of 265 sec.

For `Covertype` we performed classification using the LIBLINEAR support vector machine library. For all sample sizes the SVM dominated runtime cost, so Recursive RLS-Nyström was only marginally slower than uniform Nyström and random Fourier features for a fixed sample size.

In terms of classification performance for `Covertype` and RMSE error for `YearPredictionMSD`, as can be seen in Figure 6, both Nyström methods outperform random features when using the same number of features. However, we do not see much difference between the two Nyström methods. We leave open understanding why the significantly better kernel approximations discussed in Section 5.1 do not necessarily translate to much better learning performance, or whether they would make a larger difference for other problems.

## Footnotes

[2]Note that in Step 5 we run RECURSIVERLS-NYSTRÖM with failure probability $\delta/3$