[Reviews · NeurIPS 2017]

Reviewer 1



The authors provide an algorithm which learns a provably accurate low-rank approximation to a PSD matrix in sublinear time. Specifically, it learns an approximation that has low additive error with high probability by sampling columns from the matrix according to a certain importance measure, then forming a Nystrom approximation using these kernels. The importance measure used is an estimate of the ridge leverage scores, which are expensive to compute exactly ( O(n^3) ). Their algorithm recursively estimates these leverage score by starting from a set of columns, using those to estimate the leverage scores, sampling a set of columns according to those probabilities, and repeating ... the authors show that when this process is done carefully, the leverage score estimates are accurate enough that they can be used to get almost as good an approximation as using the true ridge leverage scores. The cost of producing the final approximation is O(ns^2) computation time and O(ns) computations of entries in the PSD matrix. This is the first algorithm which allows touching only a linear number of entries in a PSD matrix to obtain a provably good approximation --- previous methods with strong guarantees independent of the properties of the PSD matrix required forming all n^2 entries in the PSD matrix. The experimental results show that the method provides more accurate kernel approximations than Nystrom approximations formed using uniform column samples, and kernel approximations formed using Random Fourier Features. These latter two are currently the most widely used randomized low-rank kernel approximations in ML, as they are cheap to compute. A heuristic modification of the algorithm brings its runtimes down to close to that of RFFs and uniform sampled Nystrom approximations, but retains the high accuracy. I see one technical problem in the paper: the proof of Lemma 5, in the display following line 528, the first inequality seems to use the fact that if W is a diagonal matrix and 0 <= W <= Id in the semidefinite sense, then for any PSD matrix A, we have WAW <= A. This is not true, as can be seen by taking A to be the rank one outer product of [1;-1], W to be diag(1, 1/2), and x = [1, 1]. Then x^TWAWx > x^tAx = 0. Please clarify the proof if the first inequality holds for some other reason, or otherwise establish the lemma. Comments: -In section 3.2, when you introduce effective dimension, you should also cite "Learning Bounds for Kernel Regression Using Effective Data Dimensionality", Tong Zhang, Neural Computation, 2005. He calls the same expression the effective dimension of kernel methods -Reference for the claim on lines 149-150? Or consider stating it as a simple lemma for future reference -line 159, cite "On the Impact of Kernel Approximation on Learning Accuracy", Cortes et al., AISTATS, 2010 and "Efficient Non-Oblivious Randomized Reduction for Risk Minimization with Improved Excess Risk Guarantee", Xu et al., AAAI, 2017 -line 163, the reference to (13) should actually be to (5) -on line 3 of Alg 2, I suggest changing the notation of the "equals" expression slightly so that it is consistent with the definition of \tilde{l_i^lambda} in Lemma 6 -the \leq sign in the display preceding line 221 is mistyped -the experimental results with Gaussian kernels are very nice, but I think it would be useful to see how the method performs on a wider range of kernels. -would like to see "Learning Kernels with Random Features" by Sinha and Duchi referenced and compared to experimentally -would like to see "Learning Multidimensional Fourier Series With Tensor Trains", Wahls et al., GlobalSIP, 2014 cited as a related work (they optimize over frequencies to do regression versus 'random' features)

Reviewer 2



This work has two major contributions: 1. apply ridge leverage score (RLS) sampling to Nystrom method and establish several useful theoretical results, and 2. proposed an efficient and provable algorithm to approximate the ridge leverage scores. If the theories were correct, I would argue for acceptance. The proof of Theorem 8, which is the key to analyzing the efficient algorithm, seems specious. First, the proof is too terse to follow. Second, the proof seems to have errors. Third, the proof looks like induction, but it does not follow the basic rules of induction -- induction hypothesis, base case, and inductive step. The presentation of the algorithms is difficult to follow. First, Algorithm 2 is written in an incursive way. However, the function interface is undefined. Readers of this paper, including me, cannot easily implement Algorithm 2. Second, the notations are confusing. In Algorithm 2 and the proofs, the notations S0, S1, S are abused and thus confuse me. I would recommend a subscript to indicate the level of recursion. It is hard for the authors to answer my questions in short. So they'd better submit a *full* and *error-free* proof of Theorem 8. Make sure to write your mathematic induction by following a textbook example. Also make sure you do not skip any intermediate step in the proof. The authors may share me a Google Drive public link. Details regarding the proof of Theorem 8. 1. The basis step of induction hypothesis does not hold. In Algorithm 2, if S0 has 16 columns, you want to select $\sum_i p_i$ columns from the higher level which has 32 columns. However, $\sum_i p_i$ is not bounded by 32. Therefore, there won't be sufficient samples to guarantee the basis step. 2. I don't see why "... S will satisfy Eq (6) ..." Your argument just ensured one level of recursion, provided that it were correct! However, Eq (6) is the overall guarantee. 3. To establish the inductive step, you should guarantee $192 * d_eff * log (d_eff)$ be smaller than the data size in this step. It is unclear whether or not this holds. 4. Note that in each level of the recursion, since the matrix K is different, $d_eff^\lambda$ is different. You do not distinguish them. Other Comments 1. Line 157: How can additive error be "the strongest type of approximation"? Is it written in [GM13]? I think relative-error bound is. 2. Line 163: "Note that (13)". Where's (13)? 3. Do you have to describe Algorithm in the recursive way? It seems to be a bottom-up algorithm. --- after feedback --- I appreciate the authors' patient feedback. I increased my rating to weak acceptance. Two further comments: 1. In Alg 2, the set \bar{S} seems to be a set of vectors. However, in L5, it's a set of indices. 2. The proof of Thm 8 is carefully laid out. I appreciate it. However, it is very involved, and I am not confident of its correctness. The authors and readers should use their own discretion.

Reviewer 3



Recursive Sampling for the Nyström Method The paper present the concept of recursive leverage score calculation to ensure a linear time nystroem approximation without negatively effectiving the approximation accuracy. While the approach is interesting the presentation is not always sufficiently clear and also the experimental part has some missing points and could be further extended. Comments: - 'The kernel method' - there is no 'kernel method' by itself, a kernel is simple a mathemtical construct used in a number of methods leading to methods using a kernel or maybe the kernel-trick --> along this line check the remaining text to disinquish between a high dimensional feature space, a kernel function, a kernel matrix, the kernel trick and a hilbert space -- it would be quite imprecise to this all this together - 'projecting K onto a set of “landmark” data points' - acc. to your notation K is the kernel matrix this one is not actually projected onto some landmarks --> following sec 2 I see your point - but initially one expects to have K^{ns} * 1./K^{ss} * K^{sn} - 'with high probability || K -K_approx||_2 \le \lambda - really (squared) euclidean norm or not Frobenius norm? - 'using a uniform sample of 1/2 of our input points' - why 50%? - eq 1 please introduce the notation for ()^+ -- pseudo-inverse? - In Alg1 could you briefly (footnote) explain where you got the '16' - 'If S 0 has > 16 columns' - why 16 columns - this actually immediately points to the question what you typically require for N (approximating small world problems is obviously a bit useless - but at the end this is a constraint which should be mentioned in the paper) - If I got it right in Algorithm 2 ... S_0^T K S_0 ... - you expect that K (full) is available during the calculation of the algorithm? ... - this would in fact be quite unattractive in many practical cases as for e.g. N=1 Mill points it may even be impossible to provide a full kernel matrix - or very costly to calculate all the kernel function values on the fly. The nice part of the uniform landmark nystroem approach is that a full K is never needed - although by bad sampling of the landmarks the accuracy of the approximation may suffer a lot - Theorem 7 another magic constant '384' - please refer to the supplementary / or tech report / reference where the full derivation is provided (in the proof its not getting much better ' 192') - I think it is not quite clear that the derivations/proofs (and hence the algorithm) lead to lower computational costs (I am not saying your are wrong - but I think it is not clear (enough) from the provided derivation) - how many landmarks did you finally obtain / took and how many random fourier features have been used? - it would also be good to show that you approach for non-rbf kernels - e.g. a simple linear kernel representation, a polynomial kernel or other (by prob. skipping random fourier features). It would also be good to study the behaviour if the intrinsic dimension (by means of non-vanishing eigenvalues) is changing -> refering to the point what happens if the kernel space is not really very low rank - ', it never obtained high enough accuracy to be directly comparable' -- hm, if you increase the number of random fourier features - you can achieve any level of accuracy you like - so the strategy would be to increase the #features to the requested level and compare the runtime performance